# Groundwater Quality Assessment of a Multi-Layered Aquifer in a Desert Environment: A Case Study in Wadi ad-Dawasir, Saudi Arabia

**Alaa A. Masoud** [1,*] and **Ali A. Aldosari** [2]

1    Remote Sensing Laboratory, Geology Department, Faculty of Science, Tanta University, Tanta 31527, Egypt
2    Geography Department, Faculty of Arts, King Saud University, Riyadh 11451, Saudi Arabia;
     adosari@ksu.edu.sa
*    Correspondence: alaa_masoud@science.tanta.edu.eg; Tel.: +20-10-18265052

**Abstract:** Sustainable management of groundwater in desert environments dictates better knowledge of the quality status and the controlling processes. To this end, an integrated analysis of hydrochemical and statistical assessment was carried out for 692 groundwater samples collected from the multi-layered aquifer system in Wadi ad-Dawasir area (Saudi Arabia). The four water-bearing formations arranged upwards, namely Lower Wajid, Upper Wajid, Khuff-Kumdah, and Quaternary, were investigated. The prime objective was to delineate the baseline conditions and the dominant process controlling the groundwater evolution that can help make resource management better. We used fifteen indicators, namely the total dissolved solid (TDS), total hardness, Eh, pH, temperature °C, turbidity, $Fe^{2+}$, dissolved oxygen (DO), $NH_4$, $HCO_3^-$, $NO_3^-$, F, $NO_2^-$, $PO_4^{2-}$, and $SiO_2$. Descriptive statistics, violation of the international standards, geostatistical modeling, and factorial analyses (FA) were performed. Geologic, soil, topographic, and climatic factors controlling the quality were investigated. The Quaternary aquifer was the most polluted by TDS, total hardness, $NO_3^-$, $SiO_2$, $Fe^{2+}$, F, and $HCO_3^-$. Khuff-Kumdah showed largest means of DO and $NH_4$. Upper Wajid was the largest in $NO_2^-$. Lower Wajid proved largest in $PO_4^{2-}$. Violation of the international standards clarified largest emergence of the pH for the Lower Wajid; $Fe^{2+}$ and $NO_3^-$ for the Upper Wajid; and total hardness, TDS, Fluoride, turbidity, and $NH_4$ for the Quaternary aquifer. Rock interaction and evaporation are the dominant processes that contributed largely to the hydrochemical evolution of the groundwater. FA distinguished six main factors that explained for over 60.8% of the total groundwater quality variation lead byF1 (44.23%) that clarified strong positive loads of TDS (0.98), total hardness (0.95), nitrate $NO_3^-$ (0.84), turbidity (0.78), $NH_4$ (0.67), moderately loaded by fluoride (0.47), and $Fe^{2+}$ (0.31).

**Keywords:** groundwater quality indicators; geostatistical modeling; factor analysis; Wajid aquifer; Wadi ad-Dawasir; Saudi Arabia

## 1. Introduction

Agricultural sector remains the thirstiest relative to domestic water needs, which currently consume 85% of total groundwater withdrawal conducted at an unsustainable rate in water-stressed arid desert regions like Saudi Arabia [1]. However, groundwater rapid depletion at the current rates of abstraction exceeding recharge lead to water level decline and limited quantity that last less than 50 years [2] and enhanceddeterioration and hence usability has been reduced [3–6]. Quality of groundwater in these areas has been intensely influenced by the arid climate conditions (e.g., precipitation and evapotranspiration), lithology, nature of geochemical reactions, various human activities associated with overexploitation for expanding urbanization and intensified land use for agriculture activities, sewage, and industrial wastes [7–9].

To understand the hydrogeochemical parameters is necessary to utilize and protect valuable water sources effectively and predict changes in groundwater environments [10–12]. There is little knowledge about the location, extent, and type of groundwater pollution exits in the Wadi ad-Dawasir. Few hydrogeological and hydrogeochemical studies were performed in the study area [2,13–16].

Understanding the complex structure of flow systems or local pressures of natural or anthropogenic effects on the spatial hydrochemical variations can help efficient resource management of the Wadi ad-Dawasir aquifer system.

## 2. Study Area

### 2.1. Physiography and Area Characteristics

The study area covers about 8424 km$^2$ (108 km × 78 km) is located at the downstream of Wadi ad-Dawasir. The area is a green oasis and urban center developed as a desert plain for agricultural use based on the main asset of rainfall and groundwater resources (Figure 1). Cultivation is traditionally pursued in circular fields irrigated by modernized pivot sprinkling systems to fulfill water saving strategies but without any limitation. The size of the center pivot fields varies from 30 to 60 ha, where one farm can contain hundreds of fields irrigated with a number of aquifer wells. This has greatly compromised the future of non-renewable water availability for agriculture. The main crops grown in winter are wheat, potato, tomato and melon. Fodder crops, including the biennial multi-cut crops of alfalfa and Rhodes grass, are grown throughout the year; however, inactive during winter. Meteorological features of the region are speckled where the area is characterized by high temperature, high rate of evaporation, low rate of precipitation and low humidity. Diurnal temperature varies from 6 °C (winter) to 43 °C (hot summers), with an annual mean temperature of 27.4 °C. The mean annual rainfall is around 37.6 mm [17]. Topographically, the surface is partly covered by alluvium veneer sloping towards the downstream area isolated by hills and mesas that rise as much as 150 m above the plain. Water resources are under stress and groundwater levels are depleted rapidly due to heavy abstraction that may exceed crop water requirements due to high evaporation rates. The excess use of irrigational water leads to severe soil salinity problems. Alleviative measures to the predicament of the challenging water scarcity and sustainable agricultural development are urgently needed.

### 2.2. Geologic Setup

Geology and tectonic settings have been addressed in some literatures [18–23]. A platform-type basin occupied just after the cratonization of the Arabian Shield in Late Precambrian and Early Paleozoic [18] for the Wajid sandstone succession that attained long phases of non-deposition and/or erosion. The base of Wajid Sandstone lies unconformably on igneous and metamorphic rocks of the basement complex. The succession shows fining upward sequence. It is generally homogeneous, very porous, poorly cemented and interbedded with shale horizons. Large planar cross bedding is displayed throughout the sandstone horizons. The color of the rocks in different part of the section varies from white to yellow to gray-green with many red and purple hematitic bands. Fluvial environment with channel systems and unimodal paleo-current trends dominate in the southern outcrops. Shallow marine environment with littoral trace fossils in more lithologically and structurally homogenous units prevails in the north. The succession is formed by four formations.

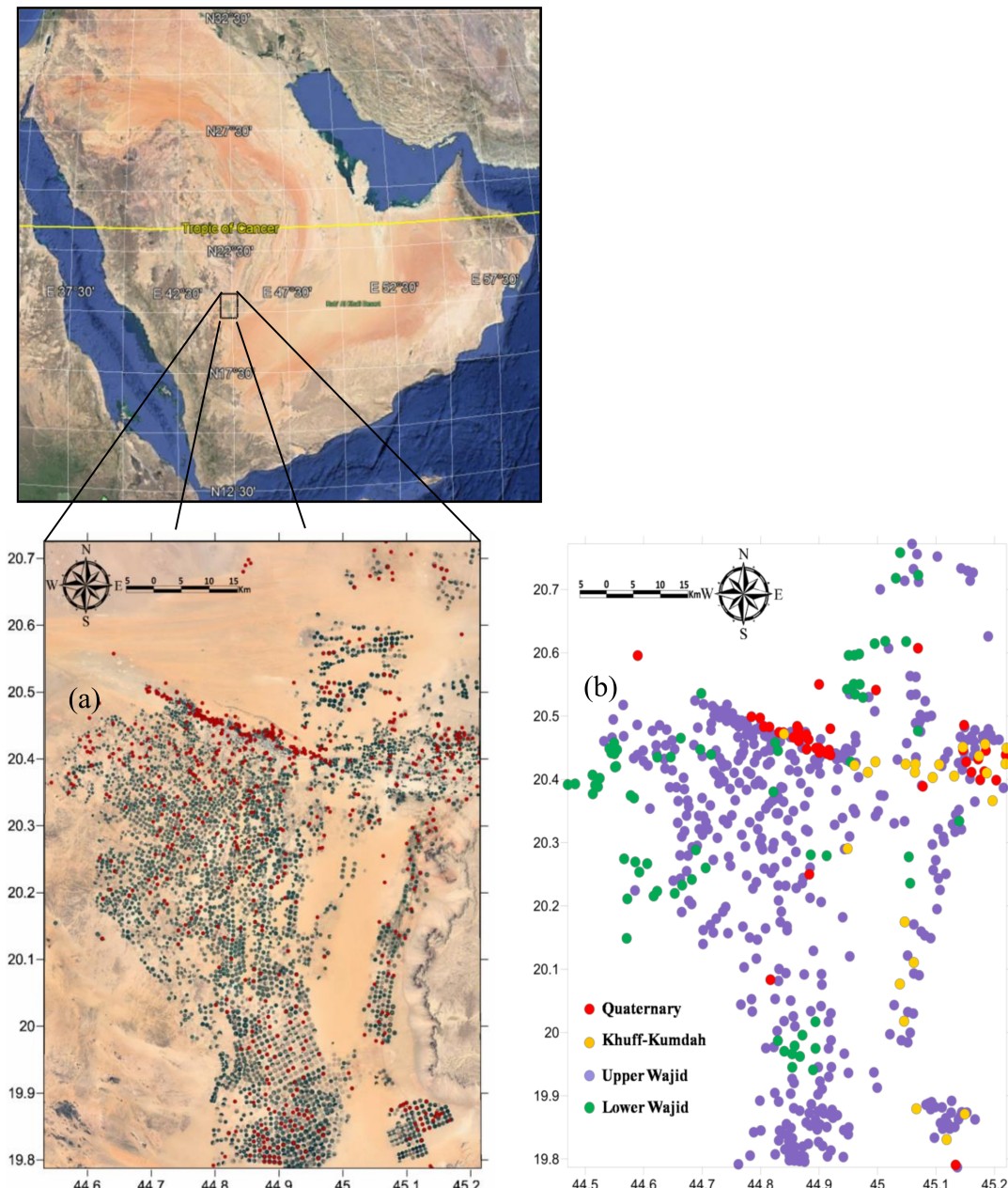

**Figure 1.** Location of sampled wells (**a**) red circles overlaying the Google earth image (December 2016), and (**b**) colored circles of the multi-layered Wajid aquifer.

During the Cambrian to Permian age (500–275 ma), Dibsiyah, Sanamah, Qalibah, Khusayyayn, and Juwayl were mostly separated by unconformity surfaces [24]. It is unconformably overlain by Khuff Formation followed by the Quaternary deposits. The Dibsiyah Formation (Fm) is a succession of medium-grained to conglomeratic sandstones with few intercalations of finer siliciclastic horizons. The Sanamah Formation is a succession of sandstones and different conglomerates that were deposited in tunnel valleys in the overall context of the Late Ordovician glaciation. The Qalibah Formation consists of two members: the Lower Qusaiba Member and the Upper Sharawra Member. The Qusaiba Member is a succession of dominantly shale with minor siltstones and sandstones; the upper unit of dominantly siltstones and sandstones is called Sharawra Member. The Khusayyayn Formation is a uniform succession of dominantly coarse sandstones deposited in medium to giant tabular foreset sets. The Juwayl Formation is the product of the Late-Paleozoic Gondwana glaciation that is unconformably overlain by the base of the Upper Permian Khuff Formation.

Six lithofacies were recognized in the Wajid Sandstone [19]. These lithofacies are identified as: silty and argillaceous sandstone, fine-grained sandstone, coarse grained sandstone; conglomerate and massive sandstone. Mineral composition suggests that the Wajid Sandstone is considered clean sands for it consists 95% of quartz grains 5% heavy minerals, mica, potash feldspar, clay matrix, ferruginous cement. The Wajid Sandstone is overlain disconformably by carbonate rocks and sandstones of the basal Khuff Formation of the Upper Permian and to the southeast, limestone of Jurassic age lie unconformably on the Wajid Sandstone. According to [24], shale of 20 cm thickness has been found in the Qalibah Formation while thick massive siltstones have been observed in the Sanamah Formation attaining several 10s of centimeters thickness. Shale and siltstone are typical products of low-energy environments (e.g., lakes) or slack water in the nearshore marine environment. Subsidence during the Permian gave rise to the deposition of platform carbonate sediments (Khuff Formation) under relatively calm conditions followed upward by the Quaternary deposits.

Integrated gravity and aeromagnetic data clarified a basement depth (thickness of sedimentary succession) varying from 600 to 1150 m, giving rise to local basins with a considerable range of aquifer thicknesses (250–700 m) as has been reported in Reference [25]. The sedimentary succession has been cut by complicated system of faults trending N–S, NNE–SSW, WNW, and NNW–SSE cutting across anticlinal blocks and strike-slip fault patterns associated with the prevailing Najd fault and the transform fault systems. Four aquifers of varying hydrochemical characteristics were distinguished; Lower Wajid, Upper Wajid, Khuff-Kumdah, and the Quaternary (Figure 2). The Wajid Fm (density of 2.33 gm cm$^{-3}$) extends from the basement surface at an average depth of about 1100 m with a gradual northward increase in thickness in the order of 380 m in the south (depth range 400–780 m) to more than 600 in the north (depth range of 580–1080 m). Variation in thickness is mostly related to the distribution of local uplifts and grabens bounded by complicated fault systems. Khuff Fm (density of 2.50 gm cm$^{-3}$) overlying the Wajid Fm ranged in thickness from 250 (depth 150–400 m) in the south to 400 m (depth range of 200–600 m) in the north. The Quaternary deposits (density of 2.10–2.4 gm cm$^{-3}$) overlie the Khuff-Kumdah Fm and reach the depth of 150 m in the south to 200 m in the north.

The regional and outcrop-scale fracture system within the Wajid Group exposures west of the study area has been studied in Reference [26]. The fractures are open and at some localities, they are sealed or coated with calcite or iron oxides of vertical to sub-vertical extensional (mode 1) type. Fracture distribution is controlled by the stratigraphic, lithological, and diagenetic variations (cementation and dissolution), bed thickness, and porosity. Thick beds of Sanamah, Khusayyayn, and Juwayl attained higher fracture spacing (small fracture density), while the thin beds of Dibsiyah showed small spacing (high fracture density). The higher the porosity, the higher is the fracture spacing with calcite-cemented sandstone displayed smaller average spacing than the clay-cemented sandstone. Highly cemented sandstone showed low values of average fracture spacing compared to the poorly cemented sandstone.

*2.3. Hydrogeology*

Some hydrogeological studies have been carried out in Wadi ad-Dawasir [2,13–16]. Major irrigation water is abstracted from the water-bearing aquifers of the Wajid Sandstone with a proven reserve of more than 30,000 million m$^3$ [13,14]. Groundwater flows to the natural downstream discharge areas of the Wadi ad-Dawasir. The annual recharge estimated at about 114 million cubic meters per year reported in the Water Atlas and Ministry of Planning [27], was extremely small, particularly in relation to the amount of stored water of 30,000 million cubic meters. However, the groundwater level has been falling in the southern part by the present because the natural discharge from the aquifer appeared to be exceeding the recharge that the aquifer could yield over 100 million cubic meters from storage with serious declines in the head or water level. Shallow wells, therefore, would have to be deepened to accommodate the decline. The low recharge and the high rate of withdrawal from wells depletes the water. Wajid Aquifer may be considered as a single hydraulic unit with fairly homogeneous lithology with fossil water—non-renewable water where it flows towards the northeast and discharge in the downstream area of Wadi ad-Dawasir.

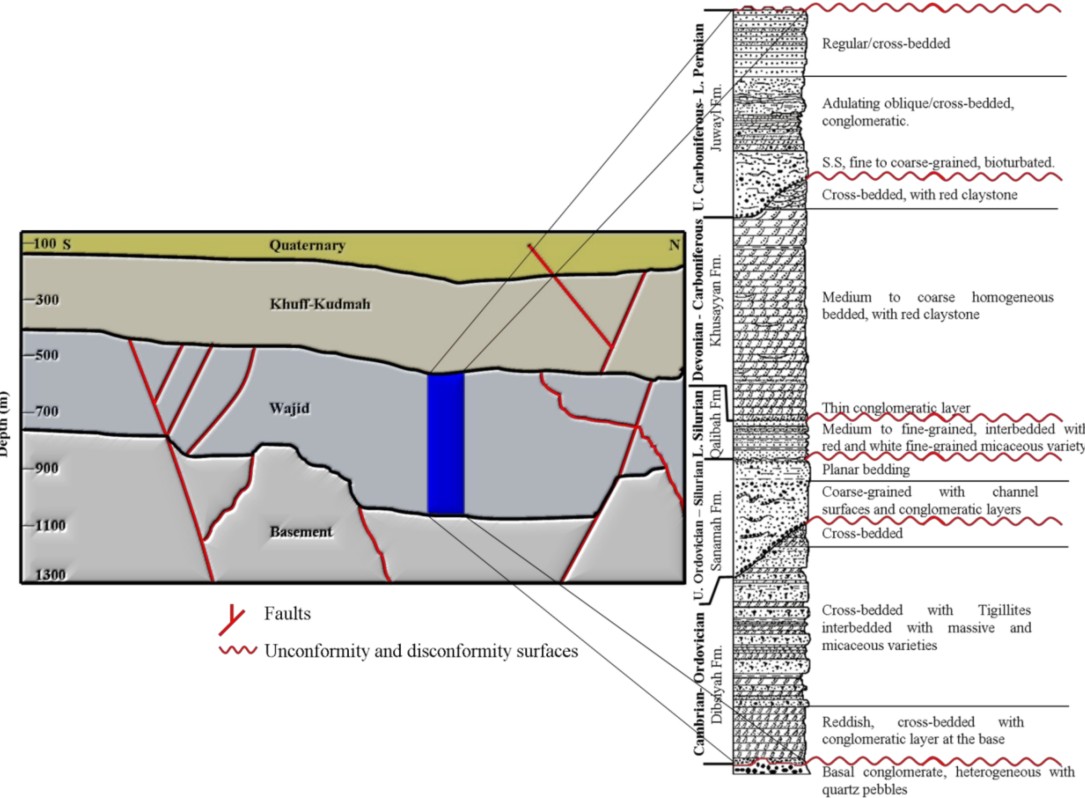

**Figure 2.** Stratigraphy and structures of the area (modified after Al Asmari et al., 2016 and Benaafi et al., 2019 [25,26]).

According to reference [15] Wajid aquifer attained average transmissivity, hydraulic conductivity, and storage coefficient of 252.8 m$^2$ day$^{-1}$, 0.421 m day$^{-1}$, and 0.0035, respectively. The discharge is much greater than the groundwater recharge leading a decline of the heads from 700 m to 550 above sea level (a.s.l.) with average recharge rate of about 15 mm year$^{-1}$ and discharge assumed from 51 wells (Q = 1800 USG/min = 9816.5 m$^3$ day$^{-1}$). Pumping tests of water levels clarified a drawdown of 43, 96, and 141 m over 60 years, with a 20-year interval [15]. In order to boost agricultural production so as to ensure a higher level of food security and an improved rural standard of living, Saudi Arabia has implemented a series of policies, since 1980, to better manage the non-sustainable water used for irrigation. An example of the best alternative was to save about 211.9 MCM from the measured water level declination of 200 m to combat for the massive abstractions occurred in 1980s in Wajid aquifer [16]. The alluvial aquifer (100–150 m thick) extends along Wadi Dawasir overlying the Khuff Formation with the recharge mainly due to upward leakage from Wajid and Khuff Formations due to high piezometeric pressure in and surrounding the faulted and fractured zone. The alluvial aquifer is discharged subsequently through excessive abstraction, and evapotranspiration.

The Wajid Aquifer of Wadi ad-Dawasir has sustained abstraction of 38 km$^3$ throughout 26 years (1983–2006), averaging just over 1.46 km$^3$ withdrawn per year. These aquifers will lose productivity (and produce brackish water) and will not be able to sustain potable and non-potable water demands projected for the next 10–30 years [2].

## 3. Materials and Methods

### 3.1. Hydrochemistry

Groundwater samples of 692 collected from the multi-aquifer system of the four water-bearing formations, namely Lower Wajid, Upper Wajid, Khuff-Kumdah, and Quaternary, in Wadi ad-Dawasir area, located 600 km SW of Riyadh city, Saudi Arabia, were chemically analyzed, and their quality was

evaluated. The number of samples analyzed varies from a variable to another within the formation and from one formation to another. The average estimates of the groundwater quality indicators from the water-bearing formations were used to characterize the groundwater of W. Dawasir area. Sampling and measurements of the physic-chemical contents of the groundwater was carried out in the period 2007–2009, through many projects reported, and archived by the Saudi Ministry of Agriculture and Water, which is known now as Saudi Ministry of Environment, Water, and Agriculture. The fifteen quality indicators used were the total dissolved solid (TDS), total hardness, Eh, pH, temperature °C, turbidity, $Fe^{2+}$, $NH_4$, $HCO_3^-$, $NO_3^-$, F, $NO_2^-$, $PO_4^{2-}$, and $SiO_2$. Descriptive statistics, violation of the local standards, and factorial analyses were performed. The Gibbs diagram [28] highlighted the processes dominated the groundwater evolution.

The ordinary kriging implemented in the geostatistical analyst of the ArcGIS9.3 package (ESRI Northeast Africa, Cairo, Egypt) is applied to produce the spatial maps of variables. The trial and error parameter selection was applied to build the semi-variograms and the best-fitted theoretical models. Minimum mean error, root mean error, and mean squared error, as well as attained root mean squared error close to unity, are considered to judge the best goodness of fit resulted in the best-fit models. These models were selected for further analysis, among which spherical was of major use.

The Pearson's correlation coefficients are calculated based on the symmetric coordinates approach to solve the problem of negative bias of the classical correlation analysis in variables that do not consider the 'hidden' influence of all other parts of a given composition [29,30]. In this approach, the new correlation coefficients are computed as weighted log-ratios of individual elements to the geometric mean of all components, which summarize the information of all pair-wise log-ratios with the individual elements.

In order to detect the hidden multivariate data structures explaining the variation of such compositional data, factor analysis is applied to emphasize few reduced principal factors controlling the geochemical processes. Since variance is related to absolute magnitude, variables with the greatest variance will have the greatest influence on the outcome, therefore considering and mixing all variables quoted in different units, simultaneously becomes erroneous in multivariate techniques [31] that can be solved by normalization and/or transformation. To understand the homogeneity of variance, the Shapiro–Wilk test is applied. The null-hypothesis of Shapiro–Wilk test is that the population is normally distributed. Thus, if the *p*-value is less than the chosen alpha level, then the null hypothesis is rejected and there is evidence that the data tested are not normally distributed. On the other hand, if the *p*-value is greater than the chosen alpha level, then the null hypothesis (that the data came from a normally distributed population) cannot be rejected (e.g., for an alpha level of 0.05, a dataset with a *p*-value of less than 0.05 rejects the null hypothesis that the data are from a normally distributed population). Factors are then listed and their score are calculated through iteration and the gradual convergence of communalities until the maximum change in the communalities is below a given threshold or when a maximum number of iterations are reached. The final communalities are then estimated and evaluated. Factor scores represent the observations coordinates on the PCA dimensions and their contributions in building the PCA axes as well as squared cosines (i.e., their representation quality on the different axes). More details on the methods are in implemented in XLSTAT (Addinsoft Inc., New York, NY, USA).

### 3.2. Land-Use Change and Hydrodynamics

Remote sensing techniques were adopted for monitoring the temporal change in the agricultural extent since 1984 to 2016. Three images with interval of 16 years have been clipped from the Google Earth and classified using maximum likelihood classifier (MLC) implemented in ENVI software (ITT VIS, Colorado, NY, USA). MLC was trained by the green areas picked up from the images. The areal coverage was calculated and interpreted to highlight the spatial distribution of the cultivated areas and their effect on the ever-increasing groundwater abstractions. Available water table data in 1969 and 2002 [15] has been appraised for understanding the hydrodynamics in the area. Water

table measurements from wells were interpolated using Kriging technique implemented in Arcgis 9.3, for 1969 and 2002, and a difference map was produced.

## 4. Results

### 4.1. Agricultural Expansion and Hydrodynamics

Agricultural expansion has been intensified reaching six-folds since 1984 starting at 551 $km^2$ (6.55%) of the area, 1780 $km^2$ (21.13%) in 2000, and 3107 $km^2$ (36.88%) in 2016 (Figure 3). Ever-expanding reclamation has been intensive on the peripheries of the old agricultural zones with new cultivation dominated extensively in the downstream area of the Wadi and the areas located northeast. Water tables for the 1969 and 2002 data showed a regional northeast flow with a drawdown decline from more than 105 m from the area's upper reaches to less than 75 m in the downstream area (Figure 4). This indicates that the outflow discharge is greater than the recharge putting the aquifer under a non-equilibrium state. The recharge comes mainly from precipitation and infiltration of runoff to stream beds or of runoff pounded on the outcrop of the aquifer estimated at about 15 mm/year. Local highs of water table decline marked the extensively reclaimed and the newly cultivated fields. The downstream area despite being over-abstracted showed lowest decline mostly compensated by the flow accumulation from precipitation and the return flow from irrigation water. Most of the groundwater is fossil—non-renewable that was recharged through infiltration from rainfall several thousand years ago during a colder and wetter time than at present and from recent irrigation return flow [15].

### 4.2. Hydrochemical Characteristics and Health Risks

Developing new policies for groundwater management rely greatly on detailed knowledge of the quality conditions of the aquifers to assure the safe services they provide. Descriptive statistics (minimum, maximum, mean, standard deviation, and counts) of the aquifers (Table 1) proved important insights for the aquifers and their pollutants. Total hardness values ranged from 92 to 4737 mg $L^{-1}$ proved very hard with average of 607 mg $L^{-1}$ due to elevated values of calcium and magnesium cations together with carbonate, bicarbonate, chloride, and sulfate anions [32] as well as for the material and texture of pathway of groundwater, as it passes through calcareous layers and limestone of the Khuff Fm and collects carbonate as well as bicarbonate ions, which later accumulate in water resources. Water with a hardness of above 200 mg $L^{-1}$ showed recent evidences for cardiovascular disorders in addition to the scale formation in the distribution network, system fouling, and increased boiling point [33,34]. Levels higher than about1000 mg $L^{-1}$ for TDS [35] make the water unpalatable and unpleasant to users due to extreme scaling in heaters, water pipes, household appliances, and boilers.

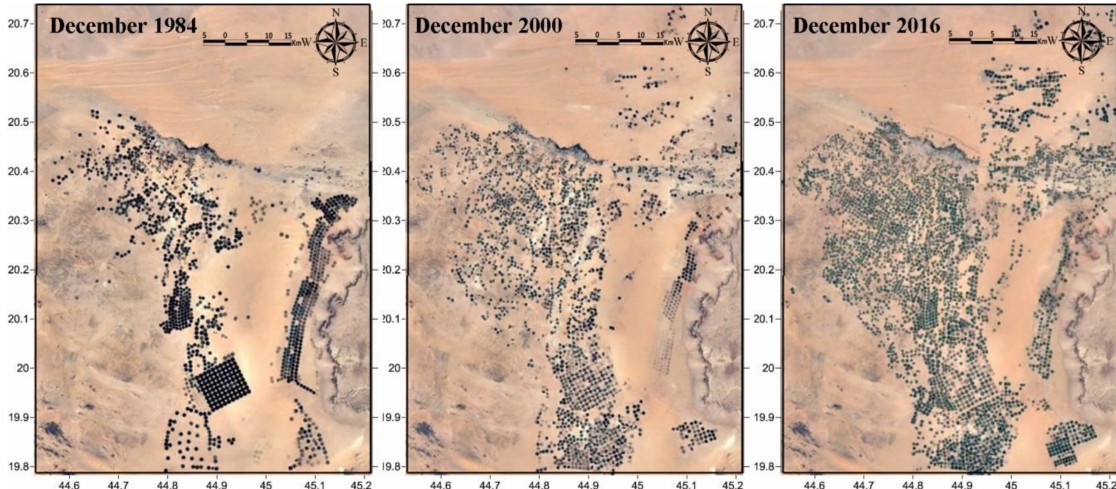

**Figure 3.** Agricultural expansion has been intensified six-folds in the 1984–2016 period.

Nitrate originates from many sources such as agricultural activities especially fertilizers, animal wastes, plant remains, industrial, and sewage disposal. In this study, its level ranged from 2.6 to 2781 mg $L^{-1}$ which suggest the influence of agricultural activities and sewage disposal. Health risks of elevated $NO_3^-$ content exceeding the maximum World Health Organization (WHO)'s allowable concentration of 50 are methaemoglobinaemia, blue baby syndrome, hypertension, diabetes, thyroid disease, stomach cancer, abortion, and altered immune function [36–38].Two zones showed the highest value of nitrate, wherein they are located in the city center, where it is densely populated and in densely cultivated areas. $Fe^{2+}$ ranged from 0.03 to 8.53 mg $L^{-1}$ with its largest average recorded in the Quaternary aquifer resulted from the widespread occurrence of ironstone and kaolinite-rich claystone lenses in the sediments.

Ammonia indicated very low concentrations reach of 0.17 to 5.66 mg $L^{-1}$, mostly owing to adsorption of clay particles and to the action of bacteria on oxidizing ammonia to nitrate and nitrite [39]. Ammonia in drinking water is not of immediate health relevance. The pH value for all the water samples is within the natural waters range of 6.5–8.5 pH for drinking and reached a maximum of 8.79 in W. Dawasir and 8.8 in Lower Wajid. Phosphate reached the maximum limit in drinking water of 0.5 mg $L^{-1}$ [40] only in the Lower Wajid samples formed mostly from fertilizer use and anthropogenic practices or from the minerals in parent rock [41,42].

The level of bicarbonate ranged from 54 to 335 mg $L^{-1}$ that lies within the WHO's standard range of 500 mg $L^{-1}$ with largest average of 177.5 marked the Quaternary. The level of soluble carbon dioxide, temperature, pH, cations, and some soluble salts control the carbonates concentration in natural waters.

Fluoride obeyed the guideline of 1.5 mg $L^{-1}$ [35] except in the Quaternary samples, where it reached a maximum of 2.98 mg $L^{-1}$ originated from rock dissolution of fluorine-rich minerals as well as from geothermal sources or from runoff, infiltration of fertilizers, industrial wastes, and manure treatment system [43,44]. In arid areas with consumption of large quantities of water, lower concentrations should be appropriate (<1 mg $L^{-1}$) [45]. Long-term exposure to fluoride in drinking water at concentrations above about 1.5 mg $L^{-1}$ can result in dental fluorosis, while values above 4 mg $L^{-1}$ can result in skeletal fluorosis and above about 10 mg $L^{-1}$, crippling fluorosis can result [46]. More than 200 million people worldwide suffer the effects of chronic endemic fluorosis, mostly in the developing countries, that are thought to be drinking water with fluoride in excess of the WHO guideline value [44].

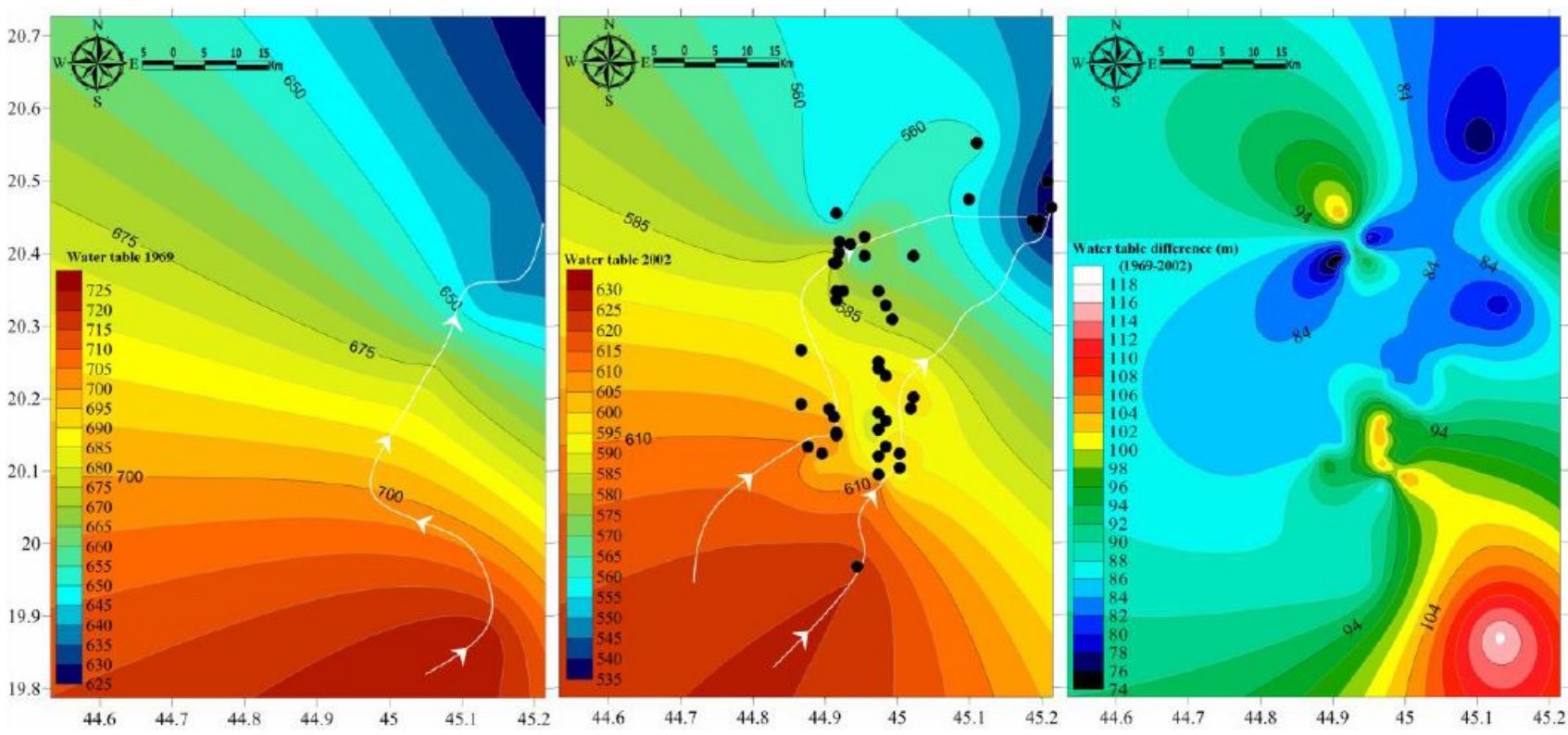

**Figure 4.** Hydrodynamics showing water table maps of the 1969, 2002, and difference with flow directions and locations of 2002 wells with water table data.

**Table 1.** Descriptive statistics of the groundwater quality indicators in Wadi ad-Dawasir.

| | | T | pH | DO | Eh | Turbidity | Total Hardness | TDS | $Fe^{2+}$ | $NH_4$ | $HCO_3^-$ | $NO_3^-$ | F | $NO_2^-$ | $PO_4^{2-}$ | $SiO_2$ |
|---|---|---|---|---|---|---|---|---|---|---|---|---|---|---|---|---|
| Quaternary | Min | 30.9 | 7.15 | 2.01 | −145 | 0.02 | 314 | 355 | 0.17 | 0.17 | 62 | 10.8 | 0.11 | 0.04 | 0.23 | 13.8 |
| | Max | 39.1 | 7.96 | 5.82 | −22 | 29.05 | 4737 | 16,500 | 8.53 | 3.54 | 335 | 2781 | 2.98 | 0.36 | 0.29 | 59.4 |
| | Mean | 32.8 | 7.5 | 4.3 | −52.9 | **2.7** | **1519** | **4659.0** | **2.8** | 1.8 | **177.0** | **483** | **0.8** | 0.1 | 0.3 | **28.0** |
| | SD | 2.1 | 0.2 | 1.0 | 42.3 | 8.3 | 1343.6 | 3734.2 | 3.3 | 1.7 | 62.1 | 952.4 | 1.0 | 0.1 | 0.0 | 12.2 |
| | count | 13 | 13 | 13 | 10 | 12 | 12 | 45 | 8 | 3 | 45 | 10 | 12 | 6 | 2 | 12 |
| Khuff-Kumdah | Min | 30.1 | 6.7 | 1.9 | −180 | 0.0 | 92 | 472 | 0.0 | 2.4 | 105 | 2.6 | 0.1 | 0.0 | 0.1 | 15.7 |
| | Max | 46 | 8.4 | 7.3 | −17 | 0.5 | 1479 | 5405 | 1.8 | 2.4 | 255 | 145.1 | 0.7 | 1.4 | 0.4 | 31.4 |
| | Mean | 32.5 | 7.6 | **4.4** | −50 | 0.1 | 629.3 | 1906.6 | 0.5 | **2.4** | 152.5 | 33.2 | 0.3 | 0.2 | 0.2 | 22.6 |
| | SD | 6.3 | 0.3 | 1.1 | 39.6 | 0.1 | 302.1 | 1061 | 0.6 | | 25 | 32.3 | 0.2 | 0.3 | 0.1 | 4.5 |
| | count | 36 | 36 | 36 | 24 | 35 | 35 | 36 | 17 | 1 | 36 | 33 | 35 | 17 | 5 | 35 |
| Upper Wajid | Min | 25 | 6.97 | 1.73 | −68 | 0.02 | 188 | 414 | 0.06 | 0 | 73 | 13.7 | 0.07 | 0.04 | 0.35 | 14.2 |
| | Max | 54.8 | 8.06 | 8.22 | 1 | 0.3 | 1387 | 3045 | 1.1 | 0 | 222 | 270 | 0.48 | 0.91 | 0.35 | 17.3 |
| | Mean | **34.4** | 7.4 | 4 | −25.5 | 0.1 | 728.1 | 1541.5 | 0.4 | | 118.5 | 71.3 | 0.2 | **0.3** | 0.4 | 15.8 |
| | SD | 4.8 | 0.3 | 1.6 | 21.5 | 0.1 | 353.1 | 670.8 | 0.4 | 0 | 41.8 | 70.8 | 0.1 | 0.5 | | 0.8 |
| | count | 119 | 11 | 11 | 8 | 11 | 11 | 46 | 6 | 0 | 24 | 11 | 9 | 3 | 1 | 11 |
| Lower Wajid | Min | 30 | 6.9 | 0.4 | −205 | 0.0 | 245 | 563 | 0.0 | 0.5 | 56 | 11.2 | 0.0 | 0.1 | 0.5 | 14.1 |
| | Max | 40.7 | 8.8 | 6.2 | 5.0 | 2.0 | 1097 | 4167 | 0.9 | 0.5 | 259 | 118 | 0.4 | 0.1 | 0.5 | 18.7 |
| | Mean | 33.6 | **7.7** | 3.8 | **−75.5** | 0.2 | 558.9 | 1518.2 | 0.4 | 0.5 | 140.7 | 52.4 | 0.1 | 0.1 | **0.5** | 16.7 |
| | SD | 3 | 0.7 | 1.3 | 84.6 | 0.5 | 243.4 | 799.7 | 0.3 | | 45.7 | 29.8 | 0.1 | 0.0 | | 1.1 |
| | count | 17 | 17 | 17 | 14 | 17 | 17 | 73 | 10 | 1 | 23 | 13 | 16 | 2 | 1 | 17 |
| W. Dawasir | Min | 3.1 | 6.63 | 0.41 | −205 | 0.02 | 92 | 83 | 0.03 | 0.05 | 54 | 2.6 | 0.04 | 0.01 | 0.01 | 10.1 |
| | Max | 54.8 | 8.79 | 13.8 | 22 | 29.05 | 4737 | 16,500 | 8.53 | 5.66 | 335 | 2781 | 5.27 | 1.38 | 16.1 | 59.4 |
| | Mean | 33.9 | 7.5 | 4.4 | −46.7 | 0.3 | 607.2 | 1838.3 | 0.9 | 1.2 | 148.5 | 68.5 | 0.3 | 0.1 | 0.8 | 18.7 |
| | SD | 4.9 | 0.4 | 1.6 | 51.9 | 2.2 | 530.0 | 1541.8 | 1.4 | 1.6 | 42.4 | 259.5 | 0.5 | 0.2 | 3.1 | 5.3 |
| | count | 185 | 186 | 175 | 125 | 179 | 179 | 686 | 104 | 16 | 238 | 153 | 173 | 78 | 26 | 178 |

Note: values in bold demarcate the largest averages in the multi-layered Wajid aquifer. TDS = total dissolved solid. T = temperature.

The Quaternary aquifer was the most affected by TDS (4659 mg L$^{-1}$), total hardness (1519 mg L$^{-1}$), NO$_3^-$ (483 mg L$^{-1}$), SiO$_2$ (28 mg L$^{-1}$), Fe$^{2+}$ (2.8 mg L$^{-1}$), F (0.8 mg L$^{-1}$), and HCO$_3^-$ (177 mg L$^{-1}$). Khuff-Kumdah showed largest means of dissolved oxygen—O$_2$ (4.4 mg L$^{-1}$) and NH$_4$ (2.4 mg L$^{-1}$), confirming its connection to the surface entered through direct absorption from the atmosphere, by rapid movement, or as a waste product of photosynthesis. Khuff-Kumdah clarified the lowest average temperature (32.5 °C) that supports easier oxygen dissolution in cooler than in warmer water. Upper Wajid clarified the largest average in NO$_2$ (0.3 mg L$^{-1}$). Lower Wajid proved largest in averages for pH (7.7), Eh (−75), and PO$_4$ (0.5 mg L$^{-1}$). Gradual upward increase was noticed for the total hardness, TDS, and Fe contents. Total hardness mean values clarified upward increase from 559 to 1519 mg L$^{-1}$ with lowered values in KhuffFm (629 mg L$^{-1}$). The salinity (average values of TDS) upward change from 1518 to 4660 mg L$^{-1}$ that implies larger contribution from marine deposits of shale in Wajid sandstone, carbonates of Khuff, and the return flow of the saline irrigation water in the Quaternary aquifer. Fe$^{2+}$ average values also increased from 0.4 to 2.8 mg L$^{-1}$ mostly related to the presence of iron-rich intercalations.

L. Wajid showed inferior groundwater quality compared to U. Wajid attributed to compositional differences. Well logs indicated the dominance of clean sandstone with high effective porosity (up to 25%), low shale content (<10%), and high water production (>75%) in the upper section while the middle and lower zones attain high shale content and hence lower porosity degrading the aquifer properties [25].

Correlation analysis indicated significant (>50%) inter-relationships among the total hardness, salinity (TDS), and the iron contents in a turbid water that is rich with NH$_4$ and NO$_3^-$ (Table 2). NH$_4$ and NO$_3^-$ strongly positively correlated with total hardness (0.90, 0.66), TDS (0.80, 0.85), Fe (0.98, 0.51), and turbidity (0.84, 0.50), respectively, and seem to be related to their elevated contents. The occurrence of these nitrogenous compounds in deep aquifers indicates that the aquifers are vertically interconnected and is connected to the surface mostly through fault systems. NH4 sourced from fertilizers and manures was also strongly related to the elevated contents of fluoride ($r = 0.66$) and silicate ($r = 0.90$) and decreases in fresh water rich with bicarbonate ($r = -0.61$) recharged from irrigation.

**Table 2.** Pearson's correlation coefficients among the studied quality indicators.

| | T | pH | DO | Eh | Turbidity | Total Hardness | TDS | Fe$^{2+}$ | NH$_4$ | HCO$_3^-$ | NO$_3^-$ | F | NO$_2^-$ | PO$_4^{2-}$ | SiO$_2$ |
|---|---|---|---|---|---|---|---|---|---|---|---|---|---|---|---|
| T | 1 | | | | | | | | | | | | | | |
| pH | 0.09 | 1 | | | | | | | | | | | | | |
| **DO** | −0.04 | 0.06 | 1 | | | | | | | | | | | | |
| Eh | −0.17 | **−0.90** | −0.09 | 1 | | | | | | | | | | | |
| Turbidity | −0.01 | −0.04 | −0.12 | −0.10 | 1 | | | | | | | | | | |
| Total Hardness | −0.21 | −0.14 | −0.09 | 0.14 | **0.57** | 1 | | | | | | | | | |
| TDS | −0.21 | −0.11 | −0.03 | 0.10 | **0.51** | 0.92 | 1 | | | | | | | | |
| Fe$^{2+}$ | 0.07 | 0.07 | −0.11 | 0.00 | 0.46 | 0.17 | 0.24 | 1 | | | | | | | |
| NH$_4$ | 0.0 | 0.05 | 0.02 | **0.49** | **0.84** | **0.90** | 0.80 | 0.98 | 1 | | | | | | |
| HCO$_3^-$ | 0.32 | 0.11 | −0.02 | −0.26 | −0.14 | −0.30 | 0.11 | −0.05 | **−0.61** | 1 | | | | | |
| NO$_3^-$ | −0.08 | −0.10 | −0.01 | 0.19 | **0.50** | 0.66 | 0.85 | 0.51 | 0.28 | −0.33 | 1 | | | | |
| F | −0.08 | −0.04 | 0.10 | −0.02 | 0.16 | 0.60 | 0.60 | 0.02 | **0.66** | −0.04 | 0.39 | 1 | | | |
| NO$_2^-$ | −0.01 | −0.02 | 0.00 | 0.07 | −0.02 | 0.06 | 0.08 | −0.08 | −0.14 | −0.24 | 0.10 | −0.02 | 1 | | |
| PO$_4^{2-}$ | −0.02 | −0.27 | −0.27 | **0.48** | −0.09 | −0.02 | −0.04 | −0.25 | −0.23 | −0.32 | 0.05 | −0.24 | 0.14 | 1 | |
| SiO$_2$ | 0.06 | 0.08 | −0.05 | −0.13 | 0.29 | 0.36 | 0.40 | 0.43 | **0.90** | 0.18 | 0.12 | 0.19 | −0.06 | −0.13 | 1 |

Note: values in bold demarcate the largest positive and negative correlation coefficients.

*4.3. Emergence and Spatial Distribution of Pollutants*

To date, this assessment has consisted primarily of physical and chemical measurements, and comparing data against world guidelines to provide an assessment of condition, as part of the water quality monitoring, evaluation and reporting program. Based on the percentage of samples exceeding the guidelines [39] (Table 3), the study area showed emergence in total hardness (98.8%), TDS (65.7%), Fe$^{2+}$ (63%), NO3− (32%), pH (5.5%), turbidity (2.85%), F (1.2%), and NH4 (0.72%), in decreasing order. All the samples were hard exceeding the level of 200 mg L$^{-1}$ except the Khuff Fm that clarified 91.4% violating the guideline. Depending on the interaction of other factors, such as pH

and alkalinity, water with hardness above approximately 200 mg $L^{-1}$ may cause scale deposition in the treatment works, distribution system and pipework and tanks within buildings.

**Table 3.** Percentage of exceedance of WHO (2011) guidelines.

|  | pH | Turbidity NTU | Total Hardness | TDS | $Fe^{2+}$ | $NH_4$ | $NO_3{}^{-}$ | F |
|---|---|---|---|---|---|---|---|---|
| **WHO 2011** | 8.5 | 1 | 200 | 1000 | 0.3 | 1.5 | 50 | 1.5 |
| Quaternary |  | **16.6** | **100** | **85.7** | 62.5 | **2.3** | 33.3 | **18.2** |
| Khuff-Kumdah |  |  | 91.4 | 70.2 | 50 |  | 14.7 |  |
| Upper Wajid | 3.4 | 1.8 | **100** | 63.4 | **68.1** | 0.5 | **33.7** |  |
| Lower Wajid | **33.3** | 6.6 | **100** | 59.7 | 44.4 |  | 33.3 |  |
| W. Dawasir | 5.5 | 2.8 | 98.8 | 65.7 | 63 | 0.7 | 32 | 1.2 |

Note: bold values demarcate the largest percentage among the studied aquifers. NTU = nephelometric turbidity units.

Violation of the international standards clarified largest emergence of the pH (33.3%) for the Lower Wajid, $Fe^{2+}$ (68.18%) and $NO_3{}^{-}$ (44.7%) for the Upper Wajid, total hardness (100%), TDS (85.7%), Fluoride (18.8%), turbidity (16.6%), and $NH_4$ (2.3%) for the Quaternary aquifer. TDS showed upward emergence with percentages of 59.7, 63.44, 70.2, and 85.7, from the Lower Wajid to the Quaternary. Largest $Fe^{2+}$ emergence was recorded for the Upper Wajid (68.18%), followed by the Quaternary (62.5%), Khuff (50%), and the Lower Wajid (44.4%). The low emergence of Lower Wajid in $Fe^{2+}$ is mostly related to the extremely low organic carbon content that favors a reducing environment [47].

As a consequence of the dominating agricultural activity (including excess application of inorganic nitrogenous fertilizers and manures), from wastewater disposal and from oxidation of nitrogenous waste products in human and animal excreta, including septic tanks, and to protect against methaemoglobinaemia in bottle-fed infants (short-term exposure), a level of 50is recommended by Reference [35]. Nitrate ($NO_3{}^{-}$) clarified largest emergence in the Upper Wajid (33.7%), followed by the Lower Wajid and Quaternary (33.3%), and the Khuff Fm (14.7%). The pH emerged at 33.3% and 3.41% for the Lower and Upper Wajid, respectively.

Turbid water is a consequence of inert clay or chalk particles or the precipitation of non-soluble reduced iron and other oxides when water is pumped from anaerobic waters and is more likely to include attached microorganisms in shallow aquifers that are a threat to health. To ensure effectiveness of disinfection, turbidity, expressed as nephelometric turbidity units (NTU), should be no more than 1 NTU, and preferably much lower [35]. Turbidity disobeying the unity dominated the Quaternary (16.6%), 6.6% of the Lower Wajid, and 1.78% for the Upper Wajid samples.

Fluoride obeyed the guidelines and emerged only at 18.18% for the Quaternary aquifer mostly concentrated from phosphatic fertilizers and dissolution of evaporative salts deposited in the arid zone during evaporation where alkalinity is greater than hardness [48]. The WHO guideline value for fluoride in drinking water is 1.5 mg $L^{-1}$. Above 1.5 mg $L^{-1}$ mottling of teeth may occur to an objectionable degree. Concentrations between 3 and 6 may cause skeletal fluorosis. Continued consumption of water with fluoride levels in excess of 10 mg $L^{-1}$ can result in crippling fluorosis.

Ammonia disobeyed the odor threshold of 1.5 mg $L^{-1}$ concentration at alkaline pH [35] for 2.3% of the Quaternary aquifer and 0.56% of the Upper Wajid samples. Nitrite ($NO_2{}^{-}$) did not present any significant concentrations and obeyed the 3 mg $L^{-1}$ limit indicating an oxidizing environment and interconnections among aquifers. No health-based guideline value is recommended for temperature, phosphates, silicates, and dissolved oxygen by Reference [35]. However, high water temperature enhances the growth of microorganisms and may increase problems related to taste, odor, color, and corrosion. Silicates and phosphates are corrosion inhibitors and they can complex dissolved iron (in the iron (II) state) and prevent its precipitation as visibly obvious red "rust". These compounds may act by masking the effects of corrosion rather than by preventing it. Orthophosphate and other phosphates are effective in suppressing dissolution of lead. Depletion of dissolved oxygen in water supplies can encourage the microbial reduction of nitrate to nitrite and sulfate to sulfide, enhances the ferrous iron content, and very high levels of dissolved oxygen may exacerbate corrosion of metal pipes.

Spatial distribution of pollutants is shown on Figure 5 where sampling points are present in the three-dimensional space. Plumes of elevated values are marked brown to red tints. TDS and total hardness plumes occur in the north western area of the four-water bearing formations. Nitrate ($NO_3^-$) was largest in the north in the Lower and Upper Wajid, and in the eastern downstream area for the Khuff and the Quaternary aquifers. Plumes of the other pollutants varied in their occurrence places within the aquifers.

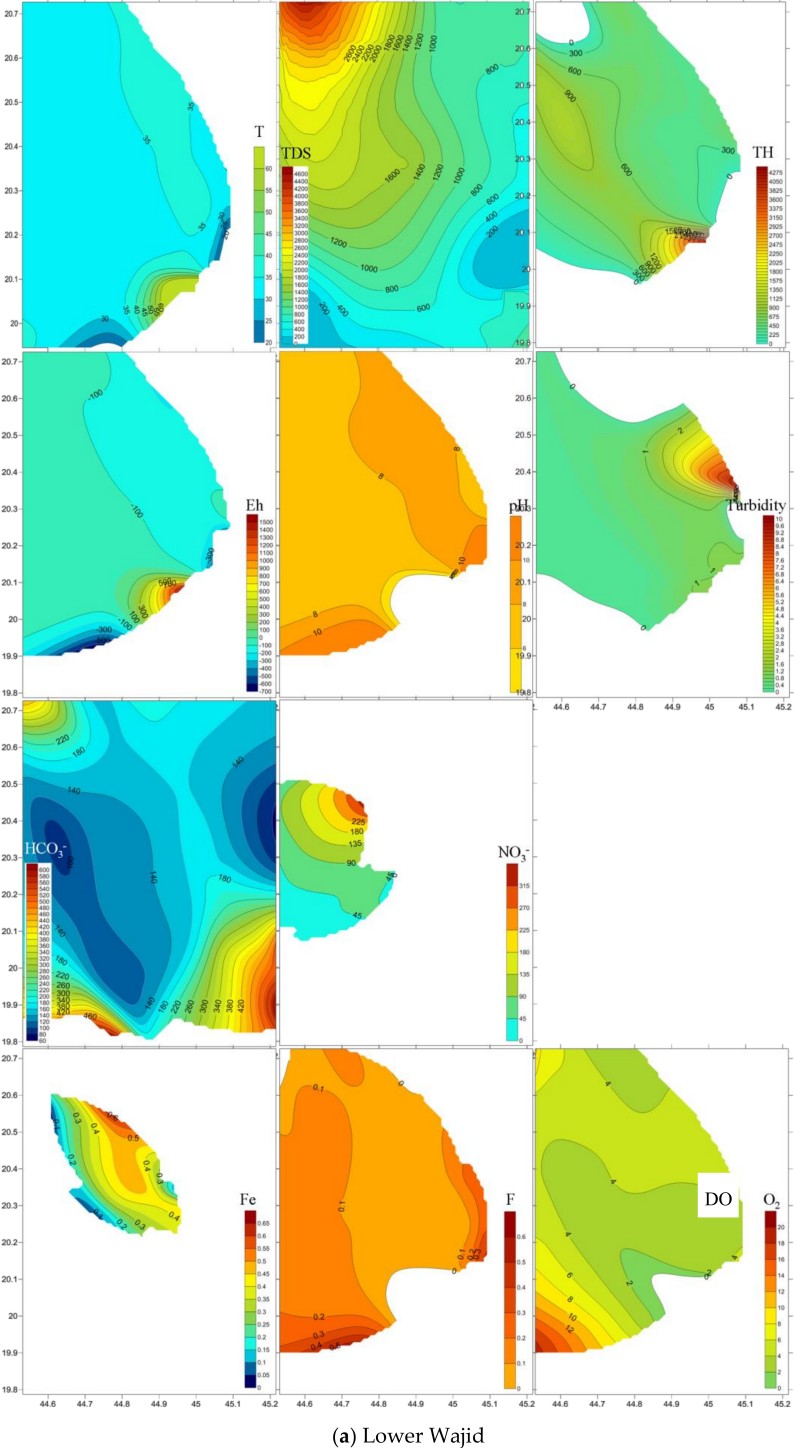

(**a**) Lower Wajid

**Figure 5.** *Cont.*

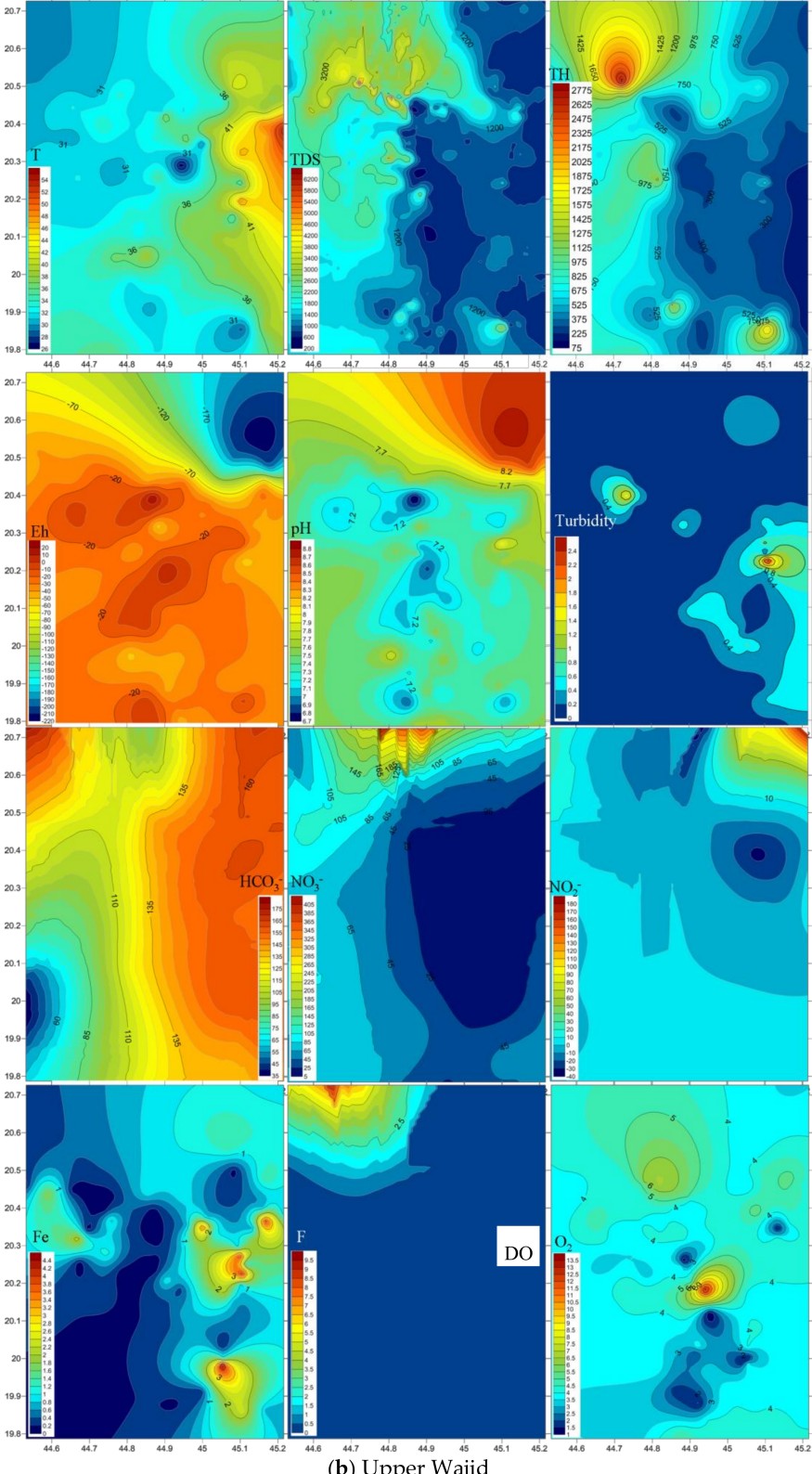

(**b**) Upper Wajid

**Figure 5.** *Cont.*

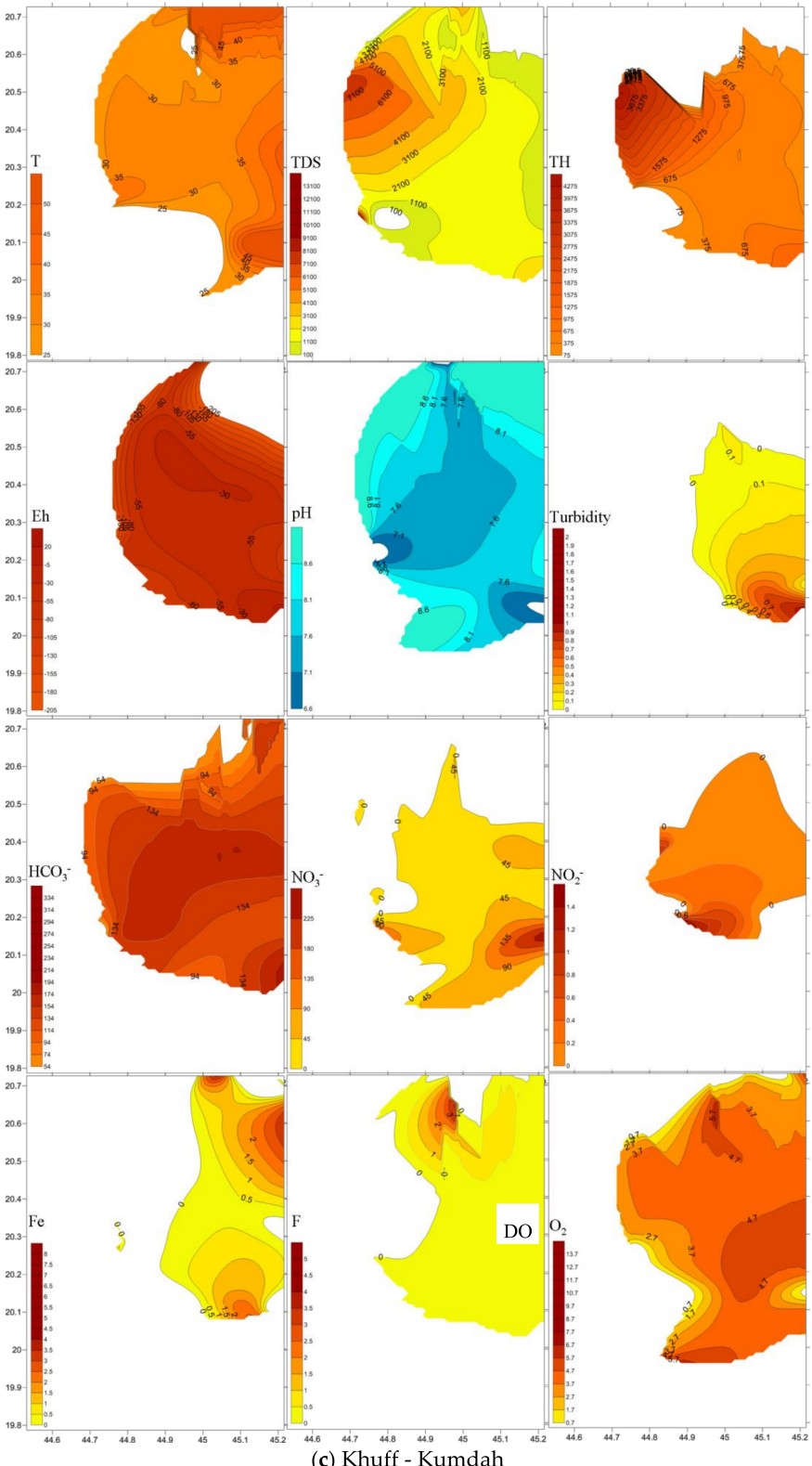

(**c**) Khuff - Kumdah

**Figure 5.** *Cont.*

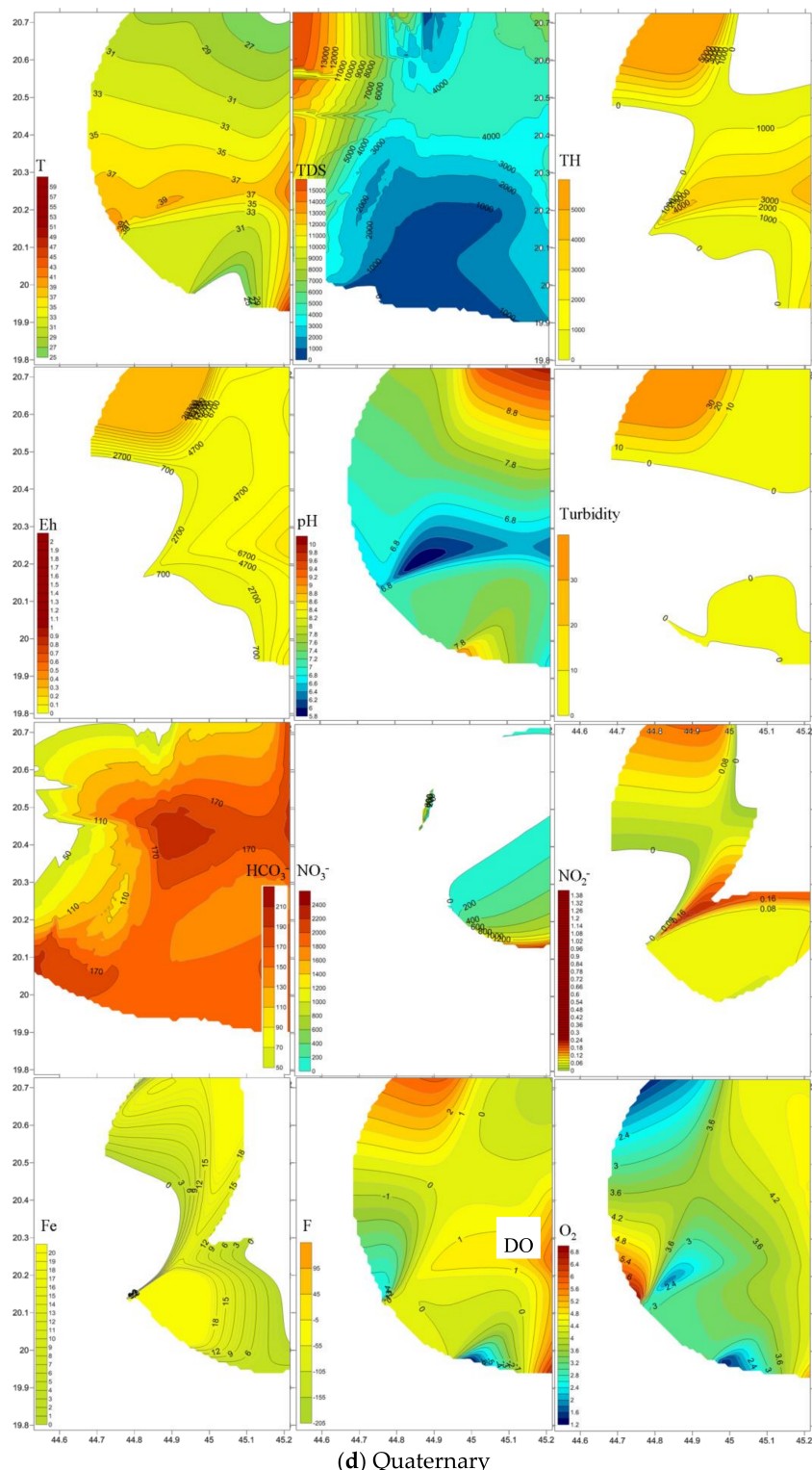

(**d**) Quaternary

**Figure 5.** Spatial distribution of pollutants in the (**a**) Lower Wajid, (**b**) Upper Wajid, (**c**) Khuff-Kumdah, and (**d**) Quaternary aquifers.

### 4.4. Mechanisms Controlling Hydrochemistry

In order to highlight the major mechanisms controlling groundwater chemistry and the dominated hydrogeochemical facies of the study area, the Gibbs diagram [28], as exhibited between TDS vs. $Na^+/(Na^+ + Ca^{2+})$ and $Cl^-/(Cl^- + HCO^-_3)$, was used and clarified the dominance of the water–rock

interaction and evaporation processes (Figure 6), primarily controlled by the chemical composition of recharge waters, water–aquifer matrix interaction, and groundwater residence time [49,50]. Vertical distribution of groundwater quality within the multi-layered aquifer showed pronounced zonation, in particular for the salinization. Water–rock interaction processes dominate at large depths (e.g., Lower and Upper Wajid) and the evaporation dominates at shallower depths (e.g., Quaternary) underpinned by the upward increase of the recharge inputs from waters influenced by intensive evaporation. Khuff-Kumdah aquifer clarified the combined dominance of the water–rock interaction and the evaporation processes.

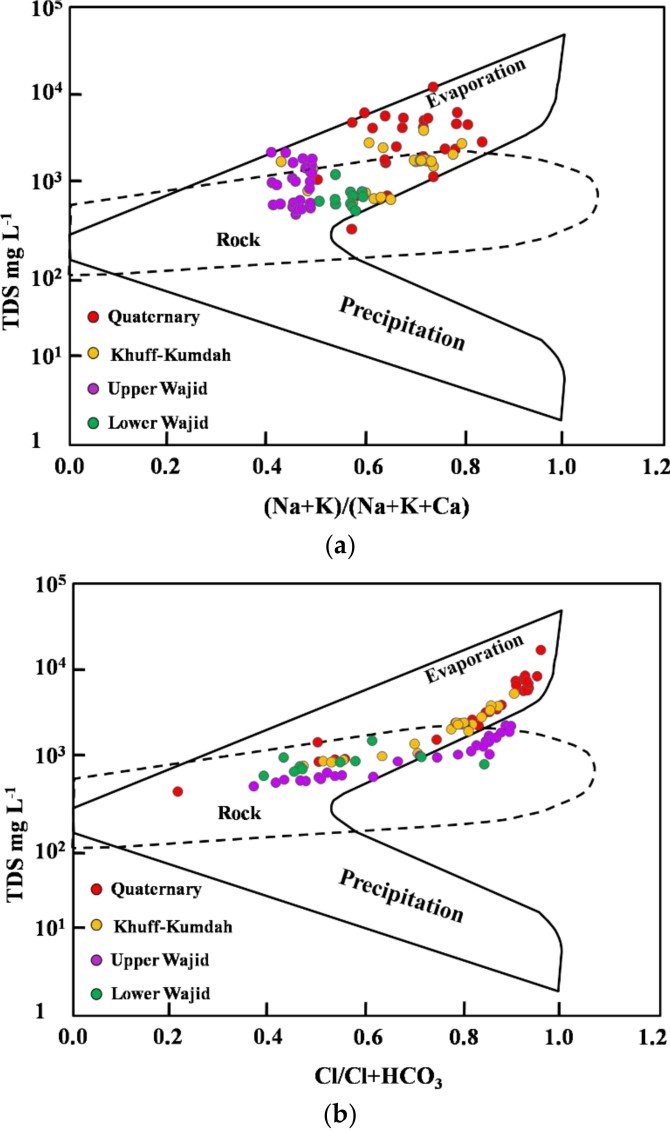

**Figure 6.** Gibbs diagrams discriminating the dominant processes of groundwater evolution by plotting the TDS (mg L$^{-1}$) against (**a**) cations (Na+K)/(Na+K+Ca) and (**b**) anions Cl/(Cl+HCO$_3$).

The excess of Na$^+$ and hence salinization result from the exchange of Ca$^{2+}$ and Mg$^{2+}$ abundant in fresh groundwater with the Na$^+$ on the surface of clay minerals, which results in an increase in Na$^+$ concentration on the expense of the decrease of the Ca$^{2+}$ and Mg$^{2+}$ concentration [51]. In addition, silicate weathering of Albite to Kaolinite in the aquifers could also contribute Na$^+$ to groundwater [52]. Dissolutions of minerals such as gypsum, anhydrite, aragonite, calcite, and dolomite are the potential ion sources to groundwater in the mineralization process [53].

In dry climatic conditions, during the long dry seasons typical for hyper-arid desert environments, areas with sparse vegetation in bare soil, shallow water table, and coarse unsaturated zone material are prone to substantial groundwater evaporation [54,55].The potential evaporation become extremely high and the recharge become practically nil after short-time precipitation events. Evaporation decreases exponentially with groundwater depth, approaching a constant value of about 0.02 mm per year for water table depths below 500 m [55].

*4.5. Factor Analysis (FA)*

To effectively utilize water resources in these arid environments, it is important to understand the spatial variability of the hydrochemical properties and the factors influencing their variations where field-derived insights are critically needed. The spatial distribution of the variables in the spaces defined by F1 and F2 is shown in Figure 7.

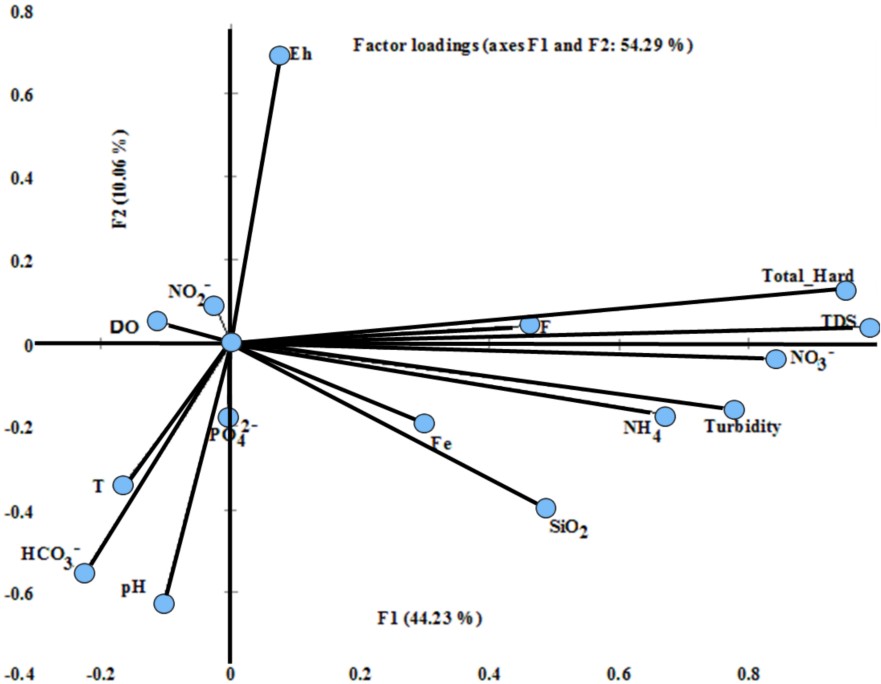

**Figure 7.** Contributions of quality parameters to groundwater data variances plotted on the loading space of primary factor (F1) versus (F2).

FA distinguished six main factors that explain for 74.34% of the total groundwater quality variation (Table 4). The shared variance (communalities) of the factors, which defined relevant to the structure was large (>0.67) for most of the variables and below this value for F, $Fe^{2+}$, T, DO, and $NO_2^-$. F1 accounting for 44.23% of the total variance had strong and positive loads of salinization indicated by the TDS (0.98), total hardness (0.95), nitrate $NO_3^-$ (0.84), turbidity (0.78), ammonia (0.67), moderately loaded by F (0.47), and $Fe^{2+}$ (0.31). F2 (10.06%) was strongly positively affected by Eh (0.70) and negatively loaded by pH (0.63), $HCO_3^-$ (0.55), and temperature T °C (0.34). Loads of $PO_4^{2-}$ (0.65) was strong on F3 (7%). F4 accounted for 6.24% and was weakly loaded by the dissolved oxygen (0.13). F5 accounted for 4.63% and was led by strong positive loads of $SiO_2$ (0.54). Negative weak loads of $NO_2^-$ (0.10) leaded F6 (2.24%). The first factor discloses the major contribution from salinization and excessive application of nitrogenous fertilizer that might be one important source of $NO_3^-$ and $NH_4$-N released by decomposition of organic matter, in addition to water–rock interaction marked by the dissemination of Fe and F. The second factor is governed by the redox dynamics and the recharge from surface mostly rainwater rich of bicarbonate. F3 reveals the extensive application of K and phosphate-rich fertilizers. F4 indicated the effect of oxygen enrichment mostly through progression of

oxygen saturation during abstraction and recovery phases [55] supported by the presence of intense network of faults and joints of the sandstone. Infiltrating oxic surface water can cause an enormous widening of the oxic zone towards the abstracting well. Silicate dissolution leads F5 while F6 clarified application of nitrogenous fertilizer as a source of the $NO_2^-$ to the groundwater pollution.

**Table 4.** Factor analysis of the studied groundwater quality parameters.

|  | F1 | F2 | F3 | F4 | F5 | F6 | Communality |
|---|---|---|---|---|---|---|---|
| T | −0.17 | **−0.34** | 0.01 | −0.14 | 0.13 | 0.14 | 0.20 |
| pH | −0.10 | **−0.63** | −0.37 | 0.40 | −0.33 | −0.12 | **0.83** |
| DO | −0.11 | 0.05 | −0.06 | **0.13** | 0.05 | 0.06 | 0.04 |
| Eh | 0.08 | **0.70** | 0.35 | −0.37 | 0.30 | −0.02 | **0.85** |
| Turbidity | **0.78** | −0.16 | −0.32 | −0.49 | −0.09 | 0.02 | **0.99** |
| Total Hardness | **0.95** | 0.13 | 0.10 | 0.18 | −0.08 | 0.07 | **0.98** |
| TDS | **0.98** | 0.04 | 0.08 | 0.16 | 0.02 | 0.00 | **1.00** |
| $Fe^{2+}$ | **0.31** | −0.20 | −0.21 | −0.26 | 0.19 | −0.18 | 0.31 |
| $NH_4$ | **0.67** | −0.17 | −0.27 | −0.43 | −0.08 | 0.08 | 0.75 |
| $HCO_3^-$ | −0.22 | **−0.55** | 0.06 | 0.00 | 0.42 | 0.37 | 0.67 |
| $NO_3^-$ | **0.84** | −0.04 | −0.27 | −0.37 | −0.19 | 0.05 | **0.96** |
| F | **0.47** | 0.04 | 0.10 | 0.33 | 0.05 | 0.29 | 0.42 |
| $NO_2^-$ | −0.02 | 0.08 | 0.02 | 0.01 | −0.07 | **−0.10** | 0.02 |
| $PO_4^{2-}$ | −0.01 | −0.18 | **0.65** | −0.27 | −0.42 | −0.08 | 0.70 |
| $SiO_2$ | 0.49 | −0.40 | 0.11 | 0.13 | **0.54** | −0.38 | **0.87** |
| Eigenvalue | 9.73 | 2.21 | 1.54 | 1.37 | 1.02 | 0.49 | |
| Variability (%) | 44.23 | 10.06 | 7 | 6.24 | 4.63 | 2.24 | |
| Cumulative% | 44.23 | 54.29 | 61.29 | 67.52 | 72.15 | 74.39 | |

Values in bold correspond for each variable to the factor for which the squared cosine is the largest.

## 5. Conclusions

The groundwater quality of the multi-layered Wajid aquifer has been assessed to delineate the baseline conditions that can help resource management and optimize the sustainability to satisfy the competing needs of socioeconomic development and maintaining healthy ecosystems in Wadi ad-Dawasir area. Expanding urbanization and intensified land use for agriculture and industrial activities has led to overexploitation of groundwater which was underpinned with rare precipitation as well as lithological compositional variations and structural controls has limited water usability. The groundwater level was declined and quality was deteriorated at varying magnitudes within the water-bearing formations. The 1969–2002 watertable data clarified a regional northeast flow with a drawdown decline from more than 105 m from the area's upper reaches to less than 75 m in the downstream area. World guidelines have been exceeded confirming emergence in the total hardness (98.8%), TDS (65.7%), $Fe^{2+}$ (63%), $NO_3^-$ (32%), pH (5.5%), turbidity (2.85%), F (1.2%), and $NH_4$ (0.72%), in decreasing order.

The Quaternary aquifer was the most affected by TDS, total hardness, $NO_3^-$, $SiO_2$, $Fe^{2+}$, F, and $HCO_3^-$. Khuff-Kumdah showed largest means of dissolved oxygen and $NH_4$ confirming its connection to the surface entered through direct absorption from the atmosphere, by rapid movement, or as a waste product of photosynthesis. Upper and Lower Wajid attained the largest averages of $NO_2^-$ and $PO_4^{3-}$, respectively. Gradual upward increase was noticed for the total hardness, TDS, and $Fe^{2+}$ contents implying larger contribution from marine deposits of shale in Wajid sandstone, carbonates of Khuff-Kumdah, and the return flow of the saline irrigation water, and $Fe^{2+}$ from iron-rich intercalations.

Lower Wajid showed inferior groundwater quality compared to Upper Wajid attributed to compositional differences. Well logs indicated the dominance of clean sandstone with high effective porosity (up to 25%), low shale content (<10%), and high water production (>75%) in the upper

section (U. Wajid) while the middle and lower zones (L. Wajid) attain high shale content and hence lower porosity degrading the aquifer properties.Groundwater chemistry of the deep aquifers (Lower and Upper Wajid) is controlled by the water–rock interaction through mineral dissolution and cation exchange, as a result of longer residence times in the aquifers and shows a trend evolving from freshwater to brackish water. The shallow Quaternary phreatic water is affected by intensive evaporation which accelerates evaporated salt dissolution in the downstream area during flood periods and the infiltration of large amounts of fresh rain water, and therefore, these waters are much saltier than the deep water. The Khuff-Kumdah aquifer exhibited the combined dominance of the rock and evaporation processes.

The statistical analyses were justified applying the symmetric coordinates approach to solve the problem of negative bias of the classical correlation analysis and compositional data were normally distributed to address and explain the factors controlling the geochemical processes.

Results showed the effective contribution rock alteration and evaporation dominated the geochemical evolution of the deep and the shallow aquifers, respectively. Results also recommend the abstraction from the Upper Wajid aquifer with considered measures to prevent or ameliorate health risks from the elevated total hardness, $NO_3^-$, and $Fe^{2+}$. It is also recommended to restrict the agriculture crops to low water demanding products. Further, new well drillings should consider the cone of depression to not to interfere, and reduce well drilling to avoid exacerbation of salinization and water level decline, and to conduct a periodic water quality assessment and solute transport modelling to project future deterioration.

Results of this research can provide fundamental information for coping with future issues such as water conflicts and climate warming and also provide references for understanding the hydrogeological processes in similar aquifers elsewhere in the world.

**Author Contributions:** Conceptualization, A.A.M. and A.A.A.; methodology, A.A.M.; software, A.A.M.; validation, A.A.M.; formal analysis, A.A.M.; investigation, A.A.A.; resources, A.A.A.; data curation, A.A.A.; writing—original draft preparation, A.A.M.; writing—review and editing, A.A.M.; visualization, A.A.M.; project administration, A.A.A.; funding acquisition, A.A.A. All authors have read and agreed to the published version of the manuscript.

**Funding:** This research was funded by the Deanship of Scientific Research at King Saud University, grant number RGP-VPP-275.

**Acknowledgments:** The authors would like to extend their sincere appreciation to the Deanship of Scientific Research at King Saud University for its funding of this research through the Research Group Project no. RGP-VPP-275.

**Conflicts of Interest:** The authors declare no conflict of interest.

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
