# Peer review of "Groundwater Quality Assessment of a Multi-Layered Aquifer in a Desert Environment: A Case Study in Wadi ad-Dawasir, Saudi Arabia"

_water, doi:10.3390/w12113020_

Round 1

Reviewer 1 Report

This paper addresses an important problem of sustainable management of groundwater in desert environments by applying statistical methodologies that, although simple and non-innovative, allow to easily obtain good (interpretable and useful) results. This paper should be published, but it needs a major revision in order to be acceptable for publication. Certainly the work effort was important, however, the analyses are not always clear and completed.

In the Abstract:

Lines 18 and 19: the authors do not mention the dissolved oxygen- O2. Is it 14 or 15 variables under analysis? Because Table 1 and 2 present 15 variables. So, there is some confusion regarding the number of variables under analysis.

Still regarding the dissolved oxygen- O2, sometimes it is mentioned as DO (Table 1, for example) and other times as O2 (Table 2, for example). The authors should harmonize the dissolved oxygen nomenclature.

The following suggestions concern language corrections (for the first dozen of lines), but this question of language should be further improved throughout the paper:

Page 1, line 17: “process controlled” should be “process controlling”

Page 1, line 27: “processes contributed” should be “processes that contributed”

Page 1, line 28: “factors explained” should be “factors that explained”

Page 1, line 35: “water needs that currently” should be “water needs, which currently”

Page 1, line 36: “consumes 85%” should be “consume 85%”

Page 1, line 38: “quantity to last less than 50” should be “quantity that last less than 50”

Page 2, line 44: “Understanding” should be “To understand”

Page 2, line 45: “Little knowledge” should be “There is little knowledge”

Page 2, paragraph lines 48-51: this paragraph needs to be rewritten.

Page 2, line 54: “(108kmx78km) located” should be “(108kmx78km) and is located”

Page 2, line 56: “based on main” should be “based on the main”

Page 2, line 56: “asset of the rainfall” should be “asset of rainfall”

Page 2, line 79: “interbeded” should be “interbedded”

Page 2, line 84: “prevail” should be”prevails”

etc.

A few important questions that should be analyzed:

Section Materials and Methods. Subsection Hydrochemistry:

Lines 181-183: “The Gibbs diagram [28] was then plotted to highlight the processes dominated the groundwater evolution”. Why didn’t the authors present both the diagram and its analysis? Otherwise, they did the analysis just to say that “The Gibbs diagram was then plotted”?

Lines 184-189: this paragraph needs to be rewritten and corrected (it contains a lot of imprecision). “In this approach, new variables (correlation coefficients)”, but correlations coefficients are not variables. The authors should deepen their knowledge on correlation analysis with symmetric balance, which allows to perform correlation analysis between coordinates which express one part of interest with respect to other parts in the composition, and also to present in the paper, for this purpose, the “new” Pearson correlation coefficient. The legend of Table 2 says: “Poisson’s correlation coefficients” – but how is that possible? What are Poisson’s correlation coefficients? Shouldn’t it be Pearson’s correlation coefficients with symmetric balances?

Lines 195-196: “To obtain the homogeneity of variance, data is normalized using Shapiro-Wilk test”. This sentence is incorrect. The Shapiro-Wilk test is not used to normalize data! The null-hypothesis of Shapiro-Wilk test is that the population is normally distributed. Thus, if the p value is less than the chosen alpha level, then the null hypothesis is rejected and there is evidence that the data tested are not normally distributed. On the other hand, if the p value is greater than the chosen alpha level, then the null hypothesis (that the data came from a normally distributed population) cannot be rejected (e.g., for an alpha level of .05, a dataset with a p value of less than .05 rejects the null hypothesis that the data are from a normally distributed population).

Lines 199-201: “Factor scores represent the observations coordinates on the PCA dimensions and their contributions in building the PCA axes as squared cosines (i.e., their representation quality on the different axes).” This is not correct and not true: “and their contributions in building the PCA axes as squared cosines”.

Table 1, lines 14-16: “The four water-bearing formations arranged upwards; Lower Wajid, Upper Wajid, Khuff-Kumdah, and Quaternary, are investigated”, but then in this same Table 1 is also analyzed W. Dawaser. Does W. Dawaser encompass all the other (I don’t think so, because isn’t 4 the total count? Is it another analysis? And why did the authors decided to analyze also W. Dawaser?

Line 254: “WHO” is referred here for the first time and so it should be: “World Health Organization (WHO)”.

Data are not well presented and the reader gets confused. For example:

  • The authors state (line13): “was carried out for 692 groundwater samples collected” and on page 6 line 174 theys say: “Groundwater samples of 492 collected”. So, what is the real number?
  • The authors should justify the analysis of W. Dawaser (in the Abstract they only refer “four water-bearing formations (line 15)”.
  • How is it possible that there is only an observation of PO42- in Lower Wajid and in Upper Wajid, and 2 observations in Quaternary? Theoretically, we can calculate means and standard deviations with two observations, but in practice it doesn’t seem correct, let alone in a scientific paper (which states that 692 groundwater samples were collected).
  • In the same water-bearing formation, why are there so different numbers of samples for the different variables? For example, in Quaternary 45 samples of TDS and 2 of PO42-. The authors should justify theses numbers.
  • There’s some confusion about whether there are 692 or 492 samples. And why this number and not another one? The sampling process and the survey methodology are not well explained and it seems they haven’t been well planned. The reader gets confused: the data were collected in the period 2007-2009 (line 178) but in each year ano? Just in one year? And the 4 water-bearing formations plus W. Dawaser: were their data collected in the same year?

The application of geostatistical modeling is far from complete. The authors do not correctly and completely present the methodology they have applied, they simply present the maps produced (by Arcgis 9.3, for 1969 and 2002), and then they present the figures without duly interpreting the figures/results. They don’t even mention the assumptions of the methodology like stationarity (anisotropy determination), variography analysis (spatial structural analysis), and the results of geostatistical modelling validation, etc. And why choosing these two years, 1969 and 2002, because in the paper the authors say (lines 177 and 178): “Sampling and measurements of the physic-chemical contents of the groundwater was carried out in the period 2007-2009”. Instead of 2002, why haven’t they analyzed some other year between 2007 and 2009? If it was not possible, they should further justify the choice of these years analyzed in the paper.

Line 261: “to 3.54 mg” should be “5.66 mg” (observed value in W. Dawaser).

Lines 264-265: “maximum of 8.79 in Lower Wajid” should be “maximum of 8.79 in W. Dawaser and 8.8 in Lower Wajid”.

There is some incoherence from line 249 to line 349, regarding the number of decimal places and rounding of the numbers presented. Sometimes, 4659.5 is written as 4659 (line 282), other times 177.5 (line283) appears in the paper as 177, etc. (there is a Great confusion/imprecision regarding rounding!). The authors should harmonize the decimal places, as well as the type of rounding. This same confusion can be seen in Table 3, where the values should be harmonized regarding decimal places and type of rounding.

Line331: “in the Upper Wajid (33.3%),” should be “in the Upper Wajid (33.7%),”

Line 340: “and emerged only at 18.8%” should be “and emerged only at 18.18%”

Lines 365-366: The authors should further justitfy how they obtained the Gibbs diagrams presented in Figure 6 (the variables involved). Why those diagrams and not others?

Figure 5 present for water-bearing formations the spatial distribution of pollutants. Why after having analyzed 15 variables (pollutants) do the authors just present the spatial distribution of 12 variables? This should be explained.

Subsection “4.5 Factor analysis (FA)” is far from complete. The authors should consult papers containing FA and try to follow the standard procedures. The reader does not know how FA was conducted: with the global data from the 4 water-bearing formations or from the 4 water-bearing formations plus the data from W. Dawaser? The authors present Figure 7 at the beginning of the subsection but do not perform any analysis, just an interpretation of that analysis.

The references section should be revised because not all the rules for referencing have been applied.

Author Response

REVIEWER 1:

This paper addresses an important problem of sustainable management of groundwater in desert environments by applying statistical methodologies that, although simple and non-innovative, allow to easily obtain good (interpretable and useful) results. This paper should be published, but it needs a major revision in order to be acceptable for publication. Certainly the work effort was important, however, the analyses are not always clear and completed.

In the Abstract:

#Lines 18 and 19: the authors do not mention the dissolved oxygen- O2. Is it 14 or 15 variables under analysis? Because Table 1 and 2 present 15 variables. So, there is some confusion regarding the number of variables under analysis.

The dissolved oxygen (DO) is added to the indicators (Line 19). The indicators are 15 variables under analysis and added to the manuscript as “Fifteen indicators included” (Line 18).

#Still regarding the dissolved oxygen- O2, sometimes it is mentioned as DO (Table 1, for example) and other times as O2 (Table 2, for example). The authors should harmonize the dissolved oxygen nomenclature.

We harmonized the dissolved oxygen nomenclature to dissolved oxygen (DO) (Line 19).

#The following suggestions concern language corrections (for the first dozen of lines), but this question of language should be further improved throughout the paper:

#Page 1, line 17: “process controlled” should be “process controlling”

We changed “process controlled” to “process controlling” in line 17.

#Page 1, line 27: “processes contributed” should be “processes that contributed”

We changed “processes contributed” to “processes that contributed” in line 27.

#Page 1, line 28: “factors explained” should be “factors that explained”

We changed “factors explained” to “factors that explained” in line 28.

#Page 1, line 35: “water needs that currently” should be “water needs, which currently”

We changed “water needs that currently” to “water needs, which currently” in line 36.

#Page 1, line 36: “consumes 85%” should be “consume 85%”

We changed “consumes 85%” to “consume 85%” in line 37.

#Page 1, line 38: “quantity to last less than 50” should be “quantity that last less than 50”

We changed “quantity to last less than 50” to “quantity that last less than 50” in line 39.

#Page 2, line 44: “Understanding” should be “To understand”

We changed “Understanding” should be “To understand” in line 45.

#Page 2, line 45: “Little knowledge” should be “There is little knowledge”

We changed “Little knowledge” should be “There is little knowledge” in lines 46-47.

#Page 2, paragraph lines 48-51: this paragraph needs to be rewritten.

We rewrote the paragraph in lines 50-52.

#Page 2, line 54: “(108kmx78km) located” should be “(108kmx78km) and is located”

We changed “(108kmx78km) located” to “(108kmx78km) and is located” in line 55.

#Page 2, line 56: “based on main” should be “based on the main”

We changed “based on main” to “based on the main”

#Page 2, line 56: “asset of the rainfall” should be “asset of rainfall”

We changed “asset of the rainfall” to “asset of rainfall” in line 57.

#Page 2, line 79: “interbeded” should be “interbedded”

We changed “interbeded” to “interbedded” in line 80.

#Page 2, line 84: “prevail” should be”prevails”

We changed “prevail” to “prevails” in line 85.

etc.

#A few important questions that should be analyzed:

Section Materials and Methods. Subsection Hydrochemistry:

#Lines 181-183: “The Gibbs diagram [28] was then plotted to highlight the processes dominated the groundwater evolution”. Why didn’t the authors present both the diagram and its analysis? Otherwise, they did the analysis just to say that “The Gibbs diagram was then plotted”?

We added “In order to highlight the major mechanisms controlling groundwater chemistry, Gibbs diagram [28] as exhibited between TDS vs. Na+ / (Na+ + Ca2+) and Cl / (Cl + HCO3), is used and clarified the dominance of the water–rock interaction and evaporation processes (Fig. 6)” in lines 389-397.

#Lines 184-189: this paragraph needs to be rewritten and corrected (it contains a lot of imprecision). “In this approach, new variables (correlation coefficients)”, but correlations coefficients are not variables. The authors should deepen their knowledge on correlation analysis with symmetric balance, which allows to perform correlation analysis between coordinates which express one part of interest with respect to other parts in the composition, and also to present in the paper, for this purpose, the “new” Pearson correlation coefficient. The legend of Table 2 says: “Poisson’s correlation coefficients” – but how is that possible? What are Poisson’s correlation coefficients? Shouldn’t it be Pearson’s correlation coefficients with symmetric balances?

We corrected the paragraph. We changed the “Poisson’s correlation coefficients” to “Pearson’s correlation coefficients” in line 193.

#Lines 195-196: “To obtain the homogeneity of variance, data is normalized using Shapiro-Wilk test”. This sentence is incorrect. The Shapiro-Wilk test is not used to normalize data! The null-hypothesis of Shapiro-Wilk test is that the population is normally distributed. Thus, if the p value is less than the chosen alpha level, then the null hypothesis is rejected and there is evidence that the data tested are not normally distributed. On the other hand, if the p value is greater than the chosen alpha level, then the null hypothesis (that the data came from a normally distributed population) cannot be rejected (e.g., for an alpha level of .05, a dataset with a p value of less than .05 rejects the null hypothesis that the data are from a normally distributed population).

Line 204-210: We rewrote paragraph to comply with the raised comment and added ” To understand the homogeneity of variance, the Shapiro-Wilk test is applied. The null-hypothesis of Shapiro-Wilk test is that the population is normally distributed. Thus, if the p value is less than the chosen alpha level, then the null hypothesis is rejected and there is evidence that the data tested are not normally distributed. On the other hand, if the p value is greater than the chosen alpha level, then the null hypothesis (that the data came from a normally distributed population) cannot be rejected (e.g., for an alpha level of .05, a dataset with a p value of less than .05 rejects the null hypothesis that the data are from a normally distributed population).”

Lines 199-201: “Factor scores represent the observations coordinates on the PCA dimensions and their contributions in building the PCA axes as squared cosines (i.e., their representation quality on the different axes).” This is not correct and not true: “and their contributions in building the PCA axes as squared cosines”.

#More details on the methods are in implemented in XLSTAT as written in lines 215-216.

#Table 1, lines 14-16: “The four water-bearing formations arranged upwards; Lower Wajid, Upper Wajid, Khuff-Kumdah, and Quaternary, are investigated”, but then in this same Table 1 is also analyzed W. Dawaser. Does W. Dawaser encompass all the other (I don’t think so, because isn’t 4 the total count? Is it another analysis? And why did the authors decided to analyze also W. Dawaser?

The four water-bearing formations are in the subsurface of W. Dawaser. The values for W. Dawaser are the average values estimated from the four water-bearing formations. This is added as “The average estimates of the groundwater quality indicators from the water-bearing formations are used to characterize the groundwater of W. Dawaser area.” in lines 178-179.

#Line 254: “WHO” is referred here for the first time and so it should be: “World Health Organization (WHO)”.

We added “World Health Organization (WHO) in line 267.

#Data are not well presented and the reader gets confused. For example:

The authors state (line13): “was carried out for 692 groundwater samples collected” and on page 6 line 174 theys say: “Groundwater samples of 492 collected”. So, what is the real number?

The real number is 692. We changed 492 to 692 in line 175.

#The authors should justify the analysis of W. Dawaser (in the Abstract they only refer “four water-bearing formations (line 15)”.

Dawaser is the whole study area and estimates of indicators are averages calculated from the four water-bearing formations as in lines 179-180.

#How is it possible that there is only an observation of PO42- in Lower Wajid and in Upper Wajid, and 2 observations in Quaternary? Theoretically, we can calculate means and standard deviations with two observations, but in practice it doesn’t seem correct, let alone in a scientific paper (which states that 692 groundwater samples were collected).

This is the analysis we got with few analyses for the PO42-.

#In the same water-bearing formation, why are there so different numbers of samples for the different variables? For example, in Quaternary 45 samples of TDS and 2 of PO42-. The authors should justify theses numbers.

We added “The number of samples analyzed varies from a variable to another within the formation and from one formation to another” in lines 178-179. For the raised example, the TDS analysis is performed only for 45 samples of the Quaternary and 2 of them are analyzed only for PO42-.

#There’s some confusion about whether there are 692 or 492 samples. And why this number and not another one? The sampling process and the survey methodology are not well explained and it seems they haven’t been well planned. The reader gets confused: the data were collected in the period 2007-2009 (line 178) but in each year ano? Just in one year? And the 4 water-bearing formations plus W. Dawaser: were their data collected in the same year?

The groundwater sampling and their analysis started since 2007 and finished in 2009 in W. Dawaser wells from the subsurface four water-bearing formations.

#The application of geostatistical modeling is far from complete. The authors do not correctly and completely present the methodology they have applied, they simply present the maps produced (by Arcgis 9.3, for 1969 and 2002), and then they present the figures without duly interpreting the figures/results. They don’t even mention the assumptions of the methodology like stationarity (anisotropy determination), variography analysis (spatial structural analysis), and the results of geostatistical modelling validation, etc. And why choosing these two years, 1969 and 2002, because in the paper the authors say (lines 177 and 178): “Sampling and measurements of the physic-chemical contents of the groundwater was carried out in the period 2007-2009”. Instead of 2002, why haven’t they analyzed some other year between 2007 and 2009? If it was not possible, they should further justify the choice of these years analyzed in the paper.

Line 188-193, we added “The ordinary kriging implemented in the geostatistical analyst of the ArcGIS9.3 package is applied to produce the spatial maps of variables. The trial and error parameter selection was applied to build the semi-variograms and the best-fitted theoretical models. Minimum mean error, root mean error, and mean squared error, and attained root mean squared error close to unity are considered to judge the best goodness of fit resulted in the best-fit models. These models were selected for further analysis among which spherical was of major use. 

Data was available only for the studied period 2007-2009 where sampling and analyses were continued within the period. The 1969 and 2002 reference water level data is used only to highlight the level decline.

#Line 261: “to 3.54 mg” should be “5.66 mg” (observed value in W. Dawaser).

We changed “to 3.54 mg” to “5.66 mg” in line 276.

#Lines 264-265: “maximum of 8.79 in Lower Wajid” should be “maximum of 8.79 in W. Dawaser and 8.8 in Lower Wajid”.

We changed “maximum of 8.79 in Lower Wajid” to “maximum of 8.79 in W. Dawaser and 8.8 in Lower Wajid” in lines 279-280.

#There is some incoherence from line 249 to line 349, regarding the number of decimal places and rounding of the numbers presented. Sometimes, 4659.5 is written as 4659 (line 282), other times 177.5 (line283) appears in the paper as 177, etc. (there is a Great confusion/imprecision regarding rounding!). The authors should harmonize the decimal places, as well as the type of rounding. This same confusion can be seen in Table 3, where the values should be harmonized regarding decimal places and type of rounding.

We adjusted the decimal places and type of rounding in tables.

#Line331: “in the Upper Wajid (33.3%),” should be “in the Upper Wajid (33.7%),”

We changed “in the Upper Wajid (33.3%),” to “in the Upper Wajid (33.7%),” in line 354.

#Line 340: “and emerged only at 18.8%” should be “and emerged only at 18.18%”

We changed “and emerged only at 18.8%” to “and emerged only at 18.18%” in line 363.

Lines 365-366: The authors should further justify how they obtained the Gibbs diagrams presented in Figure 6 (the variables involved). Why those diagrams and not others?

The diagram is of common use and also in Lines 389-397, we added the following at the beginning of section 4.4. Mechanisms controlling hydrochemistry “In order to highlight the major mechanisms controlling groundwater chemistry and the dominated hydro-geochemical facies of the study area, Gibbs diagram [28] as exhibited between TDS vs. Na+ / (Na+ + Ca2+) and Cl / (Cl + HCO3), is used and clarified the dominance of the water–rock interaction and evaporation processes (Fig. 6), primarily controlled by the chemical composition of recharge waters, water–aquifer matrix interaction and groundwater residence time [49-50]”.

#Figure 5 present for water-bearing formations the spatial distribution of pollutants. Why after having analyzed 15 variables (pollutants) do the authors just present the spatial distribution of 12 variables? This should be explained.

The rest 3 variables do not have sufficient number of sampling points to perform kriging spatial maps with accuracy.

#Subsection “4.5 Factor analysis (FA)” is far from complete. The authors should consult papers containing FA and try to follow the standard procedures. The reader does not know how FA was conducted: with the global data from the 4 water-bearing formations or from the 4 water-bearing formations plus the data from W. Dawaser? The authors present Figure 7 at the beginning of the subsection but do not perform any analysis, just an interpretation of that analysis.

FA was performed on the 4 water-bearing formations which represent the subsurface of W. Dawaser. The standard procedures are followed for the Factor analysis (FA) and references are cited for more details. Again, the data for W. Dawaser is an average estimates from the underlying four water-bearing fromations.

#The references section should be revised because not all the rules for referencing have been applied.

The references section is adjusted.

Reviewer 2 Report

In this manuscript, the groundwater quality in Wadi ad-Dawasir were investigated according to the indicators of 692 groundwater samples collected from four water-bearing formations, i.e., Lower Wajid, Upper Wajid, Khuff-Kumdah, and Quaternary. This work is meaningful and the obtained results could be of potential interest to a wide range of geohydrology and health scientists. However, the overall reading of manuscript is awkward. Many grammatical errors are presented. Accordingly, I recommend to publish this paper in Water after a major revision. Some comments are as follows:

  • There are obvious gramatical errors in the sentences in lines 12-14, 174-177, 239-246 as well as somewhere else, please confirm and revise.
  • PO42-in line 19 should be PO43-?
  • The specific analysis methods of water auality indicators should be presented and units of the indicators shown in tables should be added.
  • In Table 1, how to get the indicator values of Wadi ad-Dawasir according to those of Quaternarym Khuff-Kumdah, Upper Wajid and Lower Wajid?
  • Abbreviations presented in Table 1, Table 3 and line 296 should be explained as they first appear.

Author Response

REVIEWER 2:

In this manuscript, the groundwater quality in Wadi ad-Dawasir were investigated according to the indicators of 692 groundwater samples collected from four water-bearing formations, i.e., Lower Wajid, Upper Wajid, Khuff-Kumdah, and Quaternary. This work is meaningful and the obtained results could be of potential interest to a wide range of geohydrology and health scientists. However, the overall reading of manuscript is awkward. Many grammatical errors are presented. Accordingly, I recommend to publish this paper in Water after a major revision. Some comments are as follows:

#There are obvious gramatical errors in the sentences in lines 12-14, 174-177, 239-246 as well as somewhere else, please confirm and revise.

The grammatical errors are corrected.

#PO42-in line 19 should be PO43-?

Phosphate is measured as hydrogen phosphate ion that expressed as PO42-

#The specific analysis methods of water auality indicators should be presented and units of the indicators shown in tables should be added.

For space constrains in tables, the units are mentioned in the text.

#In Table 1, how to get the indicator values of Wadi ad-Dawasir according to those of Quaternarym Khuff-Kumdah, Upper Wajid and Lower Wajid?

The indicator values of Wadi ad-Dawasir are the average of all variables from the subsurface four water-bearing formations.

#Abbreviations presented in Table 1, Table 3 and line 296 should be explained as they first appear.

The abbreviations are explained in the manuscript text as they first appear.

Round 2

Reviewer 2 Report

The authors have seriously revised the manuscript, especially for the parts I commented. Accordingly, I believe the manuscript in present version has reached the requirement of WATER, and I sincerely suggest it be published.